# Genomic and Gene Expression Studies Helped to Define the Heterogeneity of Small-Cell Lung Cancer and Other Lung Neuroendocrine Tumors and to Identify New Therapeutic Targets

**Ugo Testa** *[ID]**, Elvira Pelosi and Germana Castelli**

Department of Oncology, Istituto Superiore di Sanità, Viale Regina Elena 299, 00161 Rome, Italy
* Correspondence: ugo.testa@iss.it

**Abstract:** Small-cell lung cancer (SCLC) is a high-grade neuroendocrine carcinoma, corresponding to about 15% of lung cancers, occurring predominantly in smokers and associated with a very poor prognosis. Key genetic alterations very frequently observed in SCLC are represented by the loss of *TP53* and *RB1*, due to mutational events or deletions; frequent amplification or overexpression of *MYC* family genes (*MYC*, *MYCL* and *MYCN*); frequent genetic alterations by mutation/deletion of *KMT2D*, RB family members p107 (*RBL1*) and p130 (*RBL2*), *PTEN*, *NOTCH* receptors and *CREBBP*. The profile of expression of specific transcription factors allowed to differentiate four subtypes of SCLC defined according to levels of ASCL1 (SCLC-A), NEUROD1 (SCLC-N), POUF23 (SCLC-P) or YAP1 (SCLC-Y). A recent study identified the subgroup SCLC-I, characterized by the expression of inflammatory/immune-related genes. Recent studies have characterized at molecular level other lung neuroendocrine tumors, including large cell neuroendocrine cancers (LCNECs) and lung carcinoids. These molecular studies have identified some therapeutic vulnerabilities that can be targeted using specific drugs and some promising biomarkers that can predict the response to this treatment. Furthermore, the introduction of immunotherapy (immune checkpoint blockade) into standard first-line treatment has led to a significant clinical benefit in a limited subset of patients.

**Keywords:** lung cancer; small-cell lung cancer; neuroendocrine lung cancer; genomic alterations; targeted therapy





## 1. Introduction

Small-cell lung cancer (SCLC) is a unique entity at clinical and histological level compared to other types of lung cancers. It corresponds to about 13% of all newly diagnosed cases of lung cancer worldwide; it was estimated that about 250,000 new SCLC cases occur worldwide each year, with about 200,000 deaths each year [1]. This cancer is usually observed in heavy smokers. SCLC is characterized by an aggressive clinical course due to the rapid development of symptoms related to intra-thoracic tumor growth and by the presence of frequent metastases, with about two-thirds of patients displaying distant metastatic disease at initial diagnosis and is rapidly fatal [1]. SCLC is a neuroendocrine cancer and must be differentiated from more rare subtypes of large-cell neuroendocrine carcinoma and neuroendocrine carcinoid (Table 1). Neuroendocrine differentiation markers, including chromogranin A, neuron-specific enolase, Neuron adhesion molecular (NCAM or CD56) and synaptophysin are expressed in SCLCs [2]. However, neuroendocrine markers are expressed also in 10% of NSCLCs and alone cannot represent a marker to differentiate SCLCs from NSCLCs. Pulmonary neuroendocrine cells are a common cell of origin for SCLC. SCLC is the deadliest type of lung cancer, being uniformly fatal and having a median survival duration of <2 years in patients with early-stage disease and <1 year in patients with metastatic disease and 5-year survival of about 5% [3,4].

**Table 1.** Most recurrent gene alterations observed in SCLC, classified as copy number alterations, gene mutations and gene fusions.

| Copy Number Alterations | Recurrent Mutations | Gene Fusions |
|:---:|:---:|:---:|
| **Gene Deletions** *CDKN2A, FHIT, RASSF1, RB1 TP53* **Gene Amplifications** *CCNET, FGFR1. IRS2, MET, MYC, MYCL, MYCN, NFIB, SOX2, SOX4* | **Cell Cycle and Apoptosis** *RB1, RBL1, RBL2, TP53, TP73* **Epigenetic Regulators** *ARID1A, ARID1B, CHD7, CREBBP, EP300, KDM6A, KMT2A, KMT2B, KMT2C, KMT2D, PBMR1, SETD2* **NOTCH Pathway** *NOTCH1, NOTCH2, NOTCH3, NOTCH4* **Receptor Kinase Signaling** *EPHa7, PIK3CA, PTEN* **Cell Adhesion-Cytoskeleton** *ALMS1, ASPM, COBL, COL4A2, COL22A1, FMN2, KIAA1211, PDE4DIP, SLIT2* | *KIAA432-JAK2 PLEKHM2-ALK PVT1-CHD7 PVT1-CCNB1IPI RFL-MYCL1 RFL-FAM132A RFL-SMAP2* |

The treatment of lung neuroendocrine tumors remains a major challenge. In this review, we highlight recent advances in SCLC research investigating molecular alterations of SCLC and of other lung neuroendocrine tumors. These studies have led to the identification of some molecular targets suitable for targeted therapy. A better understanding of the genetic and epigenetic heterogeneity may help to identify subsets of patients amenable to treatments that may improve overall survival.

## 2. Genetic Abnormalities in SCLC

The studies carried out on SCLC until 2000 have defined the main molecular abnormalities observed in these tumors and have also shown the existence of consistent differences with respect to NSCLC: (i) *TP53* is mutated in more than 90% of SCLC, compared to less than 50% of NSCLC; (ii) *RB1* is inactivated in up to 90% of SCLC, compared to only 10–15% of NSCLC; (iii) *KRAS* is rarely mutated in SCLC (<10%) but frequently mutated in NSCLC (30–40%); (iv) *CDKN2A* is rarely abnormal in SCLC but is frequently inactivated in NSCLC; (v) the deletion of 3p (14–23) at the level of the region containing the tumor suppressor *FHIT* is very frequently observed in both SCLC and NSCLC; (vi) copy number alterations are frequent in SCLC and the majority of "hot spots" for loss of heterozygosity observed in SCLC not correspond to those found in NSCLC [5].

The main genetic alterations reported in SCLC, classified as recurrent copy number alterations involving either focal gene deletions or gene amplifications, recurrent gene mutations and gene fusion events are listed in Table 1.

The availability of SCLC tumor specimens is very limited since surgical resection is performed only in a minority of these patients do not undergo surgical resection; this limitation helps to understand why the initial studies of characterization of genetic abnormalities in SCLC have been performed on single SCLC cell lines. Tobacco smoking is the main lifestyle exposure responsible for the development of lung cancer and particularly of SCLC. Smoking exerts a carcinogenic effect through numerous compounds that cause.

DNA mutation. Using massive parallel sequencing, Pleasance and coworkers have detected in a SCLC cell line 22,910 many somatic mutations (134 at the level of exome) and showed the existence of gene signatures typical of tobacco exposure [3]. Hundreds of chemical carcinogens are generated by tobacco smoking and are responsible for the generation in the lung tissue of mutations through a three-step process processes involving: (a) chemical modification of purine residues; (b) failure to repair the mutation by genome repair pathways; (c) incorrect nucleotide incorporation opposite the distorted base dur-

ing DNA replication. G > T transversions are the more frequent substitutions found in SCLC cell lines: these mutations are typically observed in cells exposed to carcinogens (polycyclic aromatic hydrocarbons) present in tobacco smoke; particularly, enrichment of G > T mutations at CpG dinucleotides was observed in SCLC cells [6]. Furthermore, G > C transversions, enriched at CpG dinucleotides, have been detected; these events occur at the level of unmethylated CpGs [6]. These mutational signatures are typically observed in cancers derived from tissues directly exposed to tobacco smoke and are largely attributable to misreplication of DNA damage caused by tobacco carcinogens [7]. Particularly, signature 4, characterized by C > A mutations is found only in cancer types in which tobacco smoking increases cancer risk and mainly those derived from epithelial cells directly exposed to tobacco smoke, such as lung cancers [7].

An integrative genome analysis carried out on 29 SCLC exomes, 2 genomes and 15 transcriptomes has provided a first view on the spectrum of genetic abnormalities present in SCLC and have shown that SCLC is characterized by a high mutation rate corresponding to 7.4 protein-changing mutations per million base pairs, markedly higher than that observed in most tumors, a phenomenon seemingly related to tobacco carcinogens [5]. SCLCs were characterized by a high number of copy number alterations (CNAs), related to both broad and focal CNAs. Particularly frequent are the almost universal deletions involving 3p and 13q (containing *RB1*) frequent gains of 3q (the region containing the gene *SOX2*), 5p and losses of 17p (containing *TP53*) [4]. Focal amplifications involved *MYCL1* and *MYCN*: these amplifications were mutually exclusive and involved a total of 16% of patients [5]. Focal amplifications affecting 8p12 involved the *FGFR1* gene; the only significant focal deletion involved the gene *FHIT* [8].

Recurrently mutated genes involve several oncogenic driver genes in SCLC, such as *TP53*, *RB1*, *PTEN*, *CREBBP*, *EP300*, *SLIT2*, *MLL*, *COBL* and *EPHA7* [5]. Inactivating *TP53* and *RB1* mutations are the most frequent mutational events occurring in SCLC; experiments of double *TP53* and *RB1* knockout in mice showed the generation of lung tumors resembling human SCLCs; these studies showed also that *TP53* and *RB1* gene inactivations are early and necessary events in the development of SCLC [8]. Three histone modifier genes *CREBBP*, *EP300* and *MLL* are frequently mutated and together, they represent the second most frequently mutated class of genes in SCLC. Three tumor suppressor genes *PTEN*, *SLIT2* and *EPHA7* are also frequently mutated [5]. Importantly, *PTEN* mutations and *FGFR1* amplifications are potentially targetable genomic alterations [8].

Mos et al. analyzed copy number alterations in 60 patient-derived SCLC cell lines and 63 primary SCLC specimens and observed recurrent amplifications of the *MYC* family genes: *MYCL1 (8%)*, *MYCN (5%)* and *MYC* (3%) [9]. Importantly, SCLC cell lines bearing amplifications of *MYC* family genes displayed a consistent sensitivity to Aurora kinase inhibition [9].

Rudin et al. reported a comprehensive genomic analysis of 36 primary SCLC specimens and observed a pattern of recurrent gene mutations and copy number alterations similar to that reported by Peifer et al. [10]. In addition, frequent mutations of several members of the *SOX* family were observed: SOX2 was amplified in 27% of primary SCLC samples. Suppression of SOX2 expression in SCLC with SOX2 amplification resulted in an inhibition of tumor growth [10]. Recent studies support an important role for SOX2 in SCLC biology. Voigt et al. have shown that in SCLC, the genetic inactivation of *RB1* is responsible for the upregulation of SOX2 expression, mediating the induction of stem/progenitor genetic programs, promoting oncogenesis [11]. SOX2 overexpression is required for tumor initiation and maintenance in SCLC [11]. In a genetically engineered mouse model of SCLC, SOX2 is critical for tumor initiation [12]. SOX2 directly regulates MYC and MYCL in the ASCL1 and NEUROD1 subtypes [9]. Another study showed that Sox2 plays a key role in the SCLC-A (ASCL1-high ASLC1) subtype: ASCL1 recruits SOX2, which promotes INSM1 and WNT11 expression; ASCL1, SOX2 and INSM1 resulted expressed in 60% of cases [13]. However, *SOX2* targets the Hippo pathway in the ASCL1-negative, YAP1-high SCLC

(SCLC-Y) subtype [12]. *SOX2* overexpression in SCLC is involved also in the mechanism of cisplatin resistance [14].

Gene fusion events are frequently observed in SCLC; most of these fusion events are intrachromosomal [13]. Recurrent gene fusion event involves *RFL* and *MYCL1* genes; the corresponding fusion protein of 466 amino acids is composed of the first 79 amino acids of RLF and the rest of MYCL1 protein, lacking its first 27 amino acids [13].

The study of copy number alterations in SCLC showed recurrent losses in the 3p and 17p chromosome regions, harboring the *FHIT*, *RSSF1* and *TP53* genes, and losses at the level of the 13q and 10p chromosome regions harboring the *RB1* and *PTEN* genes. Amplifications at the level of 1p, 2p, and 8q regions, harboring *MYCL (L-MYC)*, *MYCN (N-MYC)* and *MYC* (c-MYC) were reported. Collectively, amplifications of *MYC* family genes occur in about 50% of cases and these mutations are mutually exclusive, thus suggesting a functional redundancy among *MYC* family gene members in their contribution to SCLC development.

The spectrum of *MYC* alterations in SCLC is not limited only to gene amplifications of *MYC* gene family members, but it is extended also to other factors interacting with MYC proteins and required for their activity. MAX gene was shown to be mutated in hereditary pheochromocytoma, a tumor of neuroendocrine origin. *MAX* gene is frequently mutated (6%) and inactivated in SCLC; *MAX* mutations are mutually exclusive with gene alterations involving *MYC* and *BRG1*, an ATPase of the SWI/SNF complex [15]. BRG1 protein regulates the expression of MAX through direct binding at the level of *MAX* gene promoter. MAX and BGR1 expression are required for the activation of neuroendocrine transcription programs and for the upregulation of some *MYC* targets, such as glycolysis [12]. *BRF1* depletion resulted in the inhibition of cell proliferation, particularly pronounced in MAX-deficient SCLC [15]. The study of a cellular model of early-stage SCLC provided evidence that *MAX* deletion markedly accelerates the SCLC progression in a *RB1/TP53*⁻ mouse model; in contrast, *MAX* deletion abrogates tumorigenesis in *MYCL*-overexpressing SCLC [16]. The *MAX* deletion induced several perturbations of cellular metabolism, such as derepression of genes involved in serine and one-carbon metabolism, providing a growth advantage to SCLC cells [16]. Llabata et al. showed that *MAX* mutant SCLCs display ASCL1 or NEUROD1 or combined ASXL1/NEUROD1 features and lack *MYC* transcriptional activity [17]. *MAX*-mutant SCLCs exhibit deficient gene repression mediated by the repressive complex, ncPRC16 [17].

*MYC* activation represents a key driver of a part of SCLCs; *MYC* expression promotes SCLC development in cooperation with *RB1* and *TP53* loss in a mouse lung cancer model to promote highly aggressive and metastatic phenotype, largely mimicking human SCLC and relapsing after an initial response to chemotherapy [18]. *MYC* expression in experimental tumors was associated with a neuroendocrine-low variant phenotype of SCLC, with high expression of the NEUROD1 transcription factor and like a SCLC subset observed in human tumors [18].

The development of next generation sequencing (NGS) technology allowed an extensive characterization of the genetic alterations observed in SCLC and helped to identify genomic-derived drug targets of therapy. Ross et al. reported the wide exome sequencing of 98 cases of SCLC (23% at stage III and 74% at stage IV; 49% primary tumors and 51% metastatic tumors). This study showed that a sensitive NGS assay can be performed on formalin fixed paraffin-enbibed biopsies of SCLC patients [19]. The most recurrently altered genes were *TP53* (86% of cases, related to gene point mutations/indel and to gene sequence truncation and more rarely to gene deletion), *RB1* (54% mainly due to gene sequence truncation and gene deletion), *MLL2* gene mutations (17%, mainly due to gene sequence truncation) and *LRP1B* gene (7%, due to gene sequence truncation) [15]. The most frequently amplified genes were *RICTOR* (10%), *MYCL1* (8%) and *FGF10* (8%) [19].

In 2015, George and coworkers have performed the first comprehensive molecular characterization of SCLCs based on genome and transcriptome analysis on 100 samples (stage I–IV), in large part primary tumors. (Figure 1) In line with previous reports, SCLC genomes displayed very high mutation rates with a median value of 8.62 nonsynonymous

mutations per million base pairs [20]. Subclonal architecture was less complex in SCLC than in lung adenocarcinoma, with a three-fold lower subclonal diversity [20]. The analysis of genomic alterations showed that *TP53* and *RB1* genes were inactivated in almost all cases at the level of both alleles; interestingly, two cases displaying normal *RB1* genes, showed evidence of chromotripsis, an event determining Cyclin D1 overexpression and thus leading to RB1 deregulation through an alternative mechanism [20]. Inactivating mutational events at the level of *TP53* and *RB1* genes included mutations, translocations, homozygous deletions, hemizygous losses, losses of heterozygosity (LOH) and LOH at higher ploidy. All these findings represent strong evidence that the inactivation of both *TP53* and *RB1* genes is a fundamental and obligatory event in the development of SCLC. [20]. In addition to *TP53*, *TP73* was somatically altered by mutations and genomic rearrangements in 13% of cases. In addition to *TP53* and *RB1*, genes involved in G-protein coupled signaling, such as *KIAA1211* (17%), *COL22A1* (17%), *RGS7* (10%) and *FPR1* (6%) are recurrently mutated [20]. The transcriptome analysis showed that the majority (77%) of SCLCs show a high expression of neuroendocrine markers, such as *CHGA* (chromogranin A) and *GRP* (gastrin releasing peptide), high levels of *DLK1* (an inhibitor of NOTCH signaling) and *ASCL1* (an oncogene of neuroendocrine lineage, whose expression is inhibited by NOTCH signaling); this gene expression pattern was suggestive of low NOTCH pathway activity. In line with this observation, NOTCH family genes inactivating mutations are frequently observed (25%) in SCLCs: *NOTCH1* (14%), *NOTCH2* (4%), *NOTCH3* (6%) and *NOTCH4* (2%) [21]. According to these findings it was suggested that NOTCH signaling inactivation may contribute to SCLC development, as supported by studies performed in a SCLC mouse model, showing that activation of NOTCH signaling induced a marked reduction of tumor number and improved animal survival; furthermore, in these models, an inhibition of neuroendocrine gene expression was induced by NOTCH activation [20]. In rare cases (*KIT* 6%) SCLC tumors displayed receptor kinase mutations [20]. Furthermore, a significant proportion of SCLC displayed mutations at the level of *RB1* gene functional homologs *RBL1* (4%) and *RBL2* (7%). In addition to gene mutations, frequent focal copy-number alterations are observed, such as *TP53*, *RB1*, *CDKN2A* homozygous losses and *FHIT* losses, as well as *FGFR1*, *IRS2*, *MYC* family genes (*MYC*, *MYCN* and *MYCL1*) amplifications [16]. In conclusion, this study confirmed the complex genomic alterations present in SCLC, with a strong involvement of the cell cycle regulatory pathway and with few therapeutically targetable genetic alterations [20].

Through sequencing and resequencing of 58 primary SCLC samples Augert et al. showed frequent genetic alterations in chromatin regulators in these tumors [22]. *KMT2D (MLL2)* gene encoding the lysine methyltransferase 2D, a key regulator of transcriptional enhancer function, exhibited truncating mutations in 8% of cases [22]. *KMT2D* mutations were associated with reduced lysine methyltransferase 2 protein levels and reduced monomethylation of histone H3 lysine 4 [22]. Less frequent mutations of lysine methyltransferases were observed at the level of *KMT2C* and *KDM6A* genes [22]. In addition, mutations in other genes encoding several epigenetic factors, including genes associated with transcriptional enhancer control, such as histone acetyltransferases *CREB* binding protein gene (CREBBP) and E1A binding protein *p300* gene (EP300), and chromodomain helicase DNA binding protein 7 gene (*CHD7*) [22]. Furthermore, mutations of the chromatin remodeling gene polibromo 1 gene (*PBRM1*) were observed in 5% of cases: this gene is located at 3p21, a chromosome region frequently deleted in SCLC [22]. Gu et al. investigated 119 SCLC patients and observed *KMT2C* and *KMT2D* mutations in 12% and 19%, respectively [23]. SCLCs with *KMT2C* and *KMT2D* mutations displayed a higher tumor mutational burden than the *KMT2C* and *KMT2D* wild-type tumors (11.7 vs. 8.5 mutations/Mb) [23]. Tumor mutational burden was particularly increased in SCLCs with concomitant *KMT2D* and *TP53* mutations [23]. A recent study in a mouse model supported a role for *KMT2D* as a lung tumor suppressor whose deficiency decreases the expression of *PER2*, a regulator of multiple glycolytic genes, determining a condition of therapeutic vulnerability to glycolytic inhibitors [24].

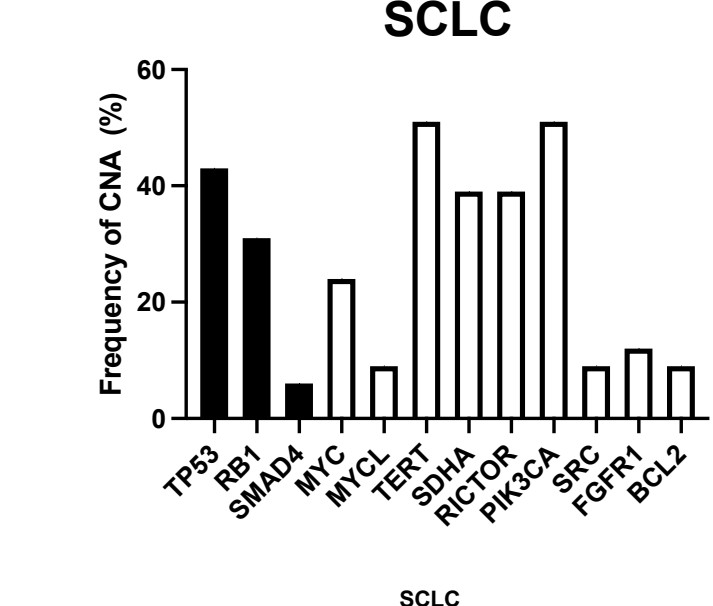

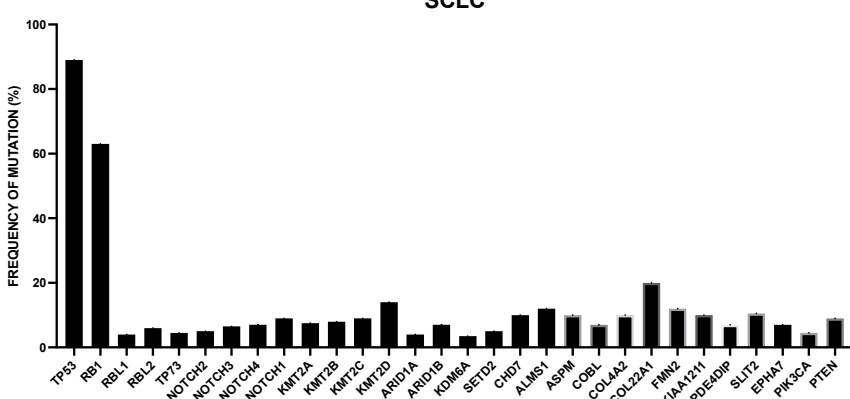

**Figure 1.** Main genetic abnormalities observed in SCLC. (**Top** Panel): Focal copy number alteration, black boxes indicate deleted genes, while white boxes indicate amplified genes. (**Bottom** Panel): recurrent gene mutations detected in SCLC patients.

As above discussed, *CREBBP*, encoding an acetyltransferase, is among the most frequently mutated genes in SCLC; the *CREBBP* mutations observed in these patients abrogate *CREBBP*-mediated histone acetylation. An autochthonous mouse model provided evidence that *CFREBBP* mutation accelerated SCLC development [25]. Gene expression studies of the experimental tumors showed that *CREBBP* loss determines a reduced expression of tight junction and cell adhesion genes, including *CDH1* [20]. Treatment with the histone deacetylase inhibitor (HDACi) Pracinostat increased histone acetylation and restored *CDH1* expression [25]. Through the study of cell lines and SCLC mouse models, Hellwig at al. further supported a higher sensitivity of *CREBBP*-mutated tumors, compared to *CREBBP*-WT tumors, to HDACis [26]. A drug sensitivity screening supported by a bioinformatic analysis showed a remarkable vulnerability to polo-like kinase 1 (PLK1) inhibition in *CREBBP*-mutant SCLC cells [27].

In a group of 39 SCLC patients with advanced disease, Dowlati and coworkers have analyzed the possible impact of molecular abnormalities on the response to therapy and observed that patients with mutant *RB1* (58% of patients) had better overall survival (11.7 months versus 9.1 months) and progression-free survival (11.2 versus 8.6 months) compared with patients with wild-type *RB1* [24]. In contrast, *TP53* mutations do not seem to have impact on progression-free survival and overall survival [28]. Udagawa et al. performed the genetic profiling of 204 SCLC patients with advanced disease at diagnosis

and reported the presence of 7% of PI3K/AKT/mTOR mutations, 75% of *TP53* mutations and 42% of *RB1* mutations [29].

As above reported, about 20–50% of SCLCs do not display *RB1* mutations. About 25% of SCLC cell lines and tumor specimens expressed RB1 protein, and seemingly represent the subgroup with WT-*RB1* [30]. Sonkin et al. explored 48 SCLC cell lines from the Cancer Cell Line Encyclopedia and observed that 8 of these cell lines were *RB1*-WT; *RB1*-WT cell lines are sensitive to CDK4/6 inhibitors [30]. Interestingly, about 4–8% of LCNECs are *RB1*-WT and display low ASCL1 and NEUROD1 expression; these tumors resemble *RB1*-WT SCLCs, an observation that has led Sonkin and coworkers to hypothesize that *RB1*-WT SCLCs and LCNECs are two faces on the same entity [31]. An ongoing clinical trial is exploring the tolerability and clinical efficacy of the CDK4/6 inhibitor Abemaciclib in *RB1*-WT SCLC and LCNEC (NCT04010357).

The presence of genetic alterations at the level of the PI3K/AKT/mTOR pathway was associated with reduced survival, whereas in limited disease patients, the presence of *TP53* mutations and the absence of *RB1* mutations were associated with unfavorable survival [25]. Next generation sequencing analysis of 26 cancer-related genes in 127 patients with SCLC undergoing treatment after surgical resection showed a frequency of 54.4% of *TP53* mutations: the presence of *TP53* mutations was associated with a relapse-free survival of 17.3 months compared to 10.4 months in the mutation-negative group [32]. Multivariate analysis showed that the presence of *TP53* mutations was a factor of prolongation of relapse-free survival but not of overall survival [32].

Recent studies have shown some remarkable differences in the genomic landscape of lung adenocarcinoma in individuals of East Asian and European ancestry [33]. Several recent studies have explored genomic profiling of SCLC in East Asians. Jiang et al. reported the whole exome sequencing and transcriptomic sequencing of 99 Chinese SCLC patients; abnormalities of *TP53* and *RB1* genes were reported in 82% and 62 of these patients, respectively. Interestingly the analysis of copy number alterations showed in 28% of patients DNA copy number gain and mRNA overexpression of Serine/Arginine Splicing Factor 1 (SRSF1), an event associated with poor survival [34]. In vitro and in vivo functional studies supported an important role of SRSF1 in tumorigenicity of SCLC; SRSF1 silencing triggers DNA-damage and suppresses PI3k/AKT and MEK/ERK pathways [34].

Hu et al. have performed a comprehensive genomic analysis in 122 Chinese SCLC patients by NGS; this analysis showed that: the most frequently altered genes were *TP53* (93%), *RB1* (79%), *LRP1B* (19%), *KTM2D* (16%), *FAT1* (11.5%), *KTM2C* (11.5%), *SPTA1* (11.5%), *STK24* (11.5%), *SAM135B* (11%), *NOTCH1* (11%) [35]. Some differences were observed between East Asian and Western patients: (i) the rate of co-occurring mutations of *TP53* and *RB1* alterations in Chinese patients was 76.6%, compared to 90.9% observed in Western patients; (ii) mutations at the level of Wnt and NOTCH signaling pathways in the Chinese SCLC patients were lower than in the Western patients; (iii) the occurrence of non-smokers among Chinese SCLC patients (29.5%) in higher than that reported in Western patients (2–10%) [36]. Another recent study reported the genomic characterization of 75 Chinese SCLC patients and reported similar frequencies of *TP53* (96%) and *RB1* (77%) mutations; in this study, frequent *SMAD4* mutations (32%) were also reported [36]. This study showed also that the median number of mutated genes per patient was 5; patients with more than 5 mutated genes and with mutated *BRCA2* gene had better progression-free survival after first-line chemotherapy than other patients [36].

A very recent study reported the genomic profiling of 50 surgically resected Chinese SCLCs, characterized by targeted deep NGS using a panel of 520 cancer-related genes. The most frequently altered genes in this cohort of patients were *TP53* (94%), *RB1* (86%), *LRP1B* (44%), *SPTA1* (26%), *KMT2D* (24%), *FAT1* (20%), *FAT3* (16%), *NOTCH2* (16%) and *ARID1A* (14%) [37]. Interestingly, *NOTCH2*, *JAK2* and *CDK12* genes were more frequently altered in Chinese patients than in European or North American SCLC patients [31]. Some genomic alterations displayed a prognostic relevance: a high tumor mutational burden (≥7 muts/Mb) were associated with a better prognosis; *ARID2* and *LRP1B* mutations were

associated with a longer overall survival; Ras signaling pathway mutations were associated with a clearly better overall survival [37].

In conclusion, although some differences were reported in one study between East Asians and European SCLC, their genomic profiling is highly comparable.

The availability of tumor tissue sufficient for high throughput sequencing at diagnosis and at relapse of SCLC patients is very limited because most of relapsed patients do not undergo tumor biopsies. Because of this limitation, only a few studies have characterized SCLCs through paired analysis at diagnosis and at relapse. Wagner et al. performed a whole-exome sequencing of paired SCLC tumor samples obtained at diagnosis and at relapse [38]. The mutational landscape of relapsed SCLC samples showed that 100% and 93% of relapse SCLC samples displayed a mutation or a deletion in *TP53* and *RB1*, respectively [38]. Interestingly, after *TP53* and *RB1*, the most frequently mutated gene was *COL11A1*, encoding for the alpha I chain of type XI collagen [38]. *COL11A1* gene dysregulation is involved in resistance to platinum chemotherapy. Several copy number alterations observed in relapsed samples are likely to play a role in mediating chemotherapy resistance, such as amplifications of *ABCC1* (a membrane ATP binding cassette transporter mediating drug export out of the cells) or deletions in mismatch repair genes *MSH2* and *MSH6*; in 10% of relapsed sample, only *MSH6* mutation were also observed [38]. Relapse samples display frequent mutations and loss of heterozygosity in modulators of WNT signaling, such as *CDH8* and *APC*. RNA sequencing studies showed a consistent enrichment for an ASCL1-low expression subtype and WNT activation in relapsed samples [32]. Increased WNT activity was associated with chemoresistance in relapsed SCLC [38].

SCLC has the strongest association with smoking among the different types of lung cancers. About 1.8% of SCLC patients are non-smokers [39]. The overall survival of never smoker SCLC patients was like that of smoker SCLC patients [39]. Analysis of genomic profiling showed that non-smoker SCLCs were characterized by lower tumor mutational burden than smoker SCLCs (1.74 vs. 8.70 mutations/Mb), lower frequency of *TP53* and *RB1* mutations, and the absence of mutational signatures related to smoking [39]. Ogino et al. have reported the genomic and characterization of 11 never/former light smokers with clinically diagnosed SCLC; in spite the clinical diagnosis of SCLC, at pathological level, the majority (8/11) displayed a mixed histology with the SCLC component associated with either non-small-cell lung cancer or atypical carcinoid or an undifferentiated carcinoma component [40]. *RB1* and *TP53* mutations were observed in 4/11 and 5/11 cases, respectively; more rarely, *EGFR*, *NRAS*, *KRAS*, *BRCA1* and *ATM* mutations were detected [40].

It is of some interest to note that SCLCs can be grouped into a central-type and a peripheral-type according to the location of the primary tumor either at the level of segmental or more proximal bronchi or sub-segmental and distal bronchi. A recent study suggested that these two SCLC types may display different molecular properties: (i) the mutational profile was similar in these two tumor types but the tumor mutational burden was higher in peripheral than in central-type tumors; (ii) amplification of 8q24.21, containing the *MYC* gene, and deletion of 13q24.21, which contains the *RB1* gene, are more frequent in peripheral-type; (iii) amplification of 12q24.31 is related to the overall survival in central but not in peripheral-type SCLC [41].

*Genomic Characterization of SCLC Using Cell-Free Tumor DNA*

The availability of tumor specimens of SCLC is limited because the majority of patients in first-line are treated directly with chemotherapy, without surgical resection. Similarly, patients in second-line are rapidly treated with chemotherapy as early as recurrence is suspected on imaging. This explains why the number of studies on the genomic landscape of SCLC patients is limited, particularly for recurrent patients. To bypass this limitation, it is of fundamental importance to find an alternative, available source of tumor cells. Recent studies have shown that some rare tumor cells circulate in the blood and release into circulation fragmented DNA, called circulating cell-free tumor DNA (ctDNA); ctDNA could represent a suitable material for the characterization of tumor genomic alterations. ctDNA

is readily detectable in the large majority of SCLC, due to the high growth rate of these tumors and to their high hematogenous spread. ctDNA may represent a precious source of to explore genomic alterations at diagnosis, to monitor response to therapy, to investigate disease evolution and to determine genetic changes at relapse. Almodovar et al. showed the feasibility of NGS evaluation of the mutational profile of SCLC patients; their evaluation of 27 patients showed that DNA mutations were detectable in 85% of patients [42]. Alongitudinal evaluation of tumor samples usually showed a reduction of VAF of DNA mutations during therapy, with an increase following completion of chemotherapy, just preceding radiological progression; similar observations were made during second-line therapy [42]. Nong et al. convincingly showed that ctDNA is an appropriate DNA source to delineate genomic landscape, subclonal architecture and to investigate genomic evolution of SCLC under therapy [43]. Importantly, a high concordance of somatic mutations between tumor DNA and ctDNA was observed [43]. A recent study by Mohan and coworkers provided evidence that it is possible to apply next generation sequencing to ctDNA derived from SCLC patients: genome-wide and targeted ctDNA sequencing identified the profile of genomic alterations in 94% of patients with limited-stage SCLC and 10% of patients with extensive-stage SCLC [43]. Both evaluation of circulating tumor cells and ctDNA readouts correlated with disease stage and overall survival [44]. Devarakonda et al. analyzed ctDNA profiling by targeted NGS in 564 SCLC patients undergoing standard treatment and samples were analyzed at diagnosis and at relapse: mean allelic frequency of gene alterations decreased from diagnosis to relapse, whereas the number of nonsynonymous mutations or amplifications detected per sample did not differ significantly following treatment and at relapse [45]. The analysis of the changes of the mutational profile at relapse suggest ctDNA profiling holds promise for evaluating mechanisms of resistance and for the identification of potential therapeutic targets, at least in some patients [45]. Feng et al. have performed a longitudinal evaluation of ctDNA profile in a cohort of 30 SCLC patients undergoing standard treatment; the patients were evaluated pre-treatment, after two, six cycles of chemotherapy and at progression [46]. A classification method was developed according to the changes in *RB1* mutational status: subtype I patients (positive at pre-treatment and after two cycles of chemotherapy) displayed an overall survival inferior to subtype II (positive at pre-treatment but negative after two cycles of chemotherapy) and subtype III (negative at both pre-treatment and after two cycles of chemotherapy) patients [46]. Furthermore, patients whose mutational tumor burden index decreased to 0 after six cycles of chemotherapy had an improved median overall survival [46].

In the peripheral blood of SCLC patients in addition to ctDNA, it is observed also the presence of CTCs. Circulating tumor cells (CTCs) are prevalent in SCLC and represent an accessible "liquid biopsy" for diagnostic, prognostic and therapeutic studies. CTCs obtained from patients with SCLC maintain their tumorigenic properties in immunocompromised mice, and the resultant CTC-derived tumor explants provide a unique tumor source for the study of the biology of SCLC and for the screening of the therapeutic response to various drugs [47]. Hou et al. have explored the number of CTCs present in 7.5 mL of blood in a cohort of SCLC patients [48]. The analysis of the data showed that the level of 50 CTCs was a suitable cutoff to distinguish the patients into a favorable and unfavorable cohort [48]. OS and PFS was significantly reduced in patients with ≥50 CTCs at baseline; in addition to the baseline values, CTC values after the first cycle of chemotherapy were prognostic [48]. Molecular analysis of CTCs of 31 patients with SCLC allowed the identification of a copy-number classifier to distinguish chemosensitive from chemorefractory patients: this classifier correctly assigned 83% of cases as chemosensitive or chemoresistant [49]. Furthermore, there was a significant difference in the progression-free survival between patients classified as chemosensitive or chemorefractory [49].

The evaluation of CTCs was prognostic also for SCLC patients with limited extension; in 60% of these patients CTCs were detectable and the best prognostic cutoff value was 15 CTC/7.5 mL of blood: patients with >15 CTCs have an OS of 5.9 months, while those with <15 CTCs display an OS of 26.7 months [50].

CTCs of SCLC represent a precious source for the isolation of tumor DNA suitable for genetic analyses. Using tumor DNA isolated from CTCs, Su et al. explored somatic mutations and copy number alterations (CNAs) by single-cell sequencing of CTCs isolated from patients undergoing chemotherapy treatment [51]. Patients with a low score of CNAs displayed an increased overall survival and a prolonged progression-free survival compared to patients with high CNA scores [52]. The analysis of CNAs at different time points during chemotherapy treatment, showed that CNA heterogeneity may derive from allelic losses of initially consistent CNAs [52].

## 3. Gene Expression Studies and Molecular Classification of SCLC

The analysis of the mutational landscape of SCLC provided important information about the major genetic drivers of tumor development but failed to define tumor subtypes. However, the analysis of the expression of specific transcription factors provided a framework to different SCLC subtypes, with specific biologic properties [53]. About 70% of SCLCs are characterized by high expression of ASCL1 and MYCLN, together with high expression of neuroendocrine markers; about 25% of NSCLCs are characterized by high expression of MYC and low expression of ASCL1 and neuroendocrine markers: these tumors are characterized by high levels of NEUROD1, POU2F3 or YAP1. Thus, it was recently proposed a classification of SCLC based on four tumor subtypes, defined each according to the high levels of a specific transcription factor: ASCL1 (SCLC-A subtype), NEUROD1 (SCLC-N subtype), POU2F3 (SCLC-P) or YAP1 (SCLC-Y) [53] (Figure 2). ASCL1 is a lineage-specific transcription factor, member of the basic helix-loop-helix (BHLH) family of transcription factors involved in neuronal commitment and neuroendocrine differentiation and reactivated in SCLC; in normal lung, ASCL1 expression is confined to quiescent progenitor neuroendocrine cells. This transcription factor is essential for tumor development and survival; ASCL1 targets *MYCL1*, *SOX2*, *BCL2*, *RET* genes and the NOTCH Ligand *DLL3*; additional important targets of ASCL1 are represented by *NKX2-1* (*TTF1*) and *BRN2* [54]. ASCL1-high tumors are associated with a high expression of neuroendocrine markers [54]. *MYCL* is amplified or highly expressed in the SCLC-A subtype and is required for SCLC-A development [40]. SCLC-N subtype is characterized by the expression of NEUROD1, a member of the NeuroD family of BHLH transcription factors, represents about 17% of all SCLCs and is characterized by a lower expression of neuroendocrine markers than SCLC-A tumors. SCLC-N tend to exhibit amplification or overexpression of MYC. In mouse models, MYC expression drives the formation of tumors with a non-neuroendocrine SCLC phenotype with NEUROD1 expression [17]. It is important to point out that the SCLC-N subtype is less defined than the SCLC-A subtype. This conclusion is strongly supported by a recent study reporting the immunohistochemical detection of ASCL1, NEUROD1, POU2F3 and YAP1 expression in 174 SCLC samples: ASCL1-only expression was observed in 41% of cases; NEUROD1-only in 8% of cases; ASCL1/NEUROD1 double-positive in 37% of cases; ASCL1/NEUROD1 double-negative in 14% of cases [55]. The distribution of ASCL1/NEUROD1 double-positive cases was established according to the dominance of the expression of one of these transcription factors over the other one; thus, although ASCL1 and NEUROD1 are frequently co-expressed, in most cases one of the two markers was strongly dominant over the other [54]. Importantly, this study showed also that both ASCL1-dominant and NEUROD1-dominant SCLC subtypes were associated with neuroendocrine marker$^{high}$/TTF1$^{high}$/DLL3$^{high}$ profile [55].

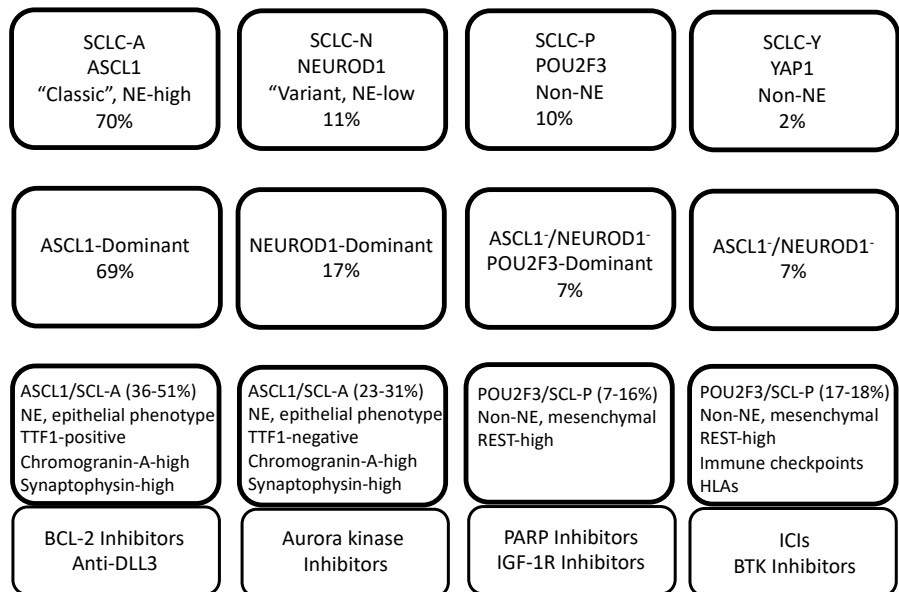

**Figure 2.** Different molecular classifications of SCLC. The first line from the top shows the model proposed by Rudin et al. in 2019 [52]; the second line the model proposed by Baine et al. in 2020 [54]; the third line the model recently proposed by Gay et al. [56]. The fourth line shows the main targeted therapies proposed for each molecular subtype.

A minority subset of ASCL1/NEUROD1-double negative SCLCs was found to express POUF3 or YAP1. A common feature of these SCLCs is represented by the absence of a neuroendocrine phenotype and expression of RE1 silencing transcription factor (REST); furthermore, these tumors display an epithelial to mesenchymal transition profile and activation of some signaling pathways, including NOTCH, HIPPO and TGFβ pathways. A part of these ASCL1/NEUROD1 double-negative SCLCs is characterized by the expression of POU2F3, a marker of chemosensory tuft cells, known in the lung airway as brush cells [55]. Immunohistochemical studies have shown that POU2F3 expression was limited to ASCL1 and NEUROD1-double negative SCLCs, of which 45% were positive for POU2F3 [55].

YAP1 has been proposed to represent the fourth SLCL subtype, associated with decreased INSM1 expression and enrichment for intact RB1 [53]. YAP1 is a modulator of transcription activated by the HIPPO signaling pathway. However, the role of YAP1 as one of the key drivers in SCLC is unclear. In fact, the recent study of immunohistochemical evaluation of ASCL1, NEUROD1, POU2F3 and YAP1 in SCLCs, YAP1 expression resulted absent or low, failing to define a distinct subtype of SCLC; the only few cases expressing moderate YAP1 levels displayed a combined histology and the most of YAP1-positive cells are not neuroendocrine cellular elements [55]. In line with these observations, Simpson et al. reported the isolation of SCLC-A, SCLC-N and SCLC-P subtypes, but not of SCLC-Y subtype in circulating tumor cell-derived xenografts [57]. Furthermore, this study showed that within the group of low/non-neuroendocrine SCLC, a rare subtype with high expression of the transcription factor ATOH1 was identified. Finally, a detailed analysis of YAP1 expression in SCLC-derived xenografts provided evidence that YAP1 expression in these tumors was observed at the level of rare non-neuroendocrine cell clusters, while some rare tumor elements with low expression of neuroendocrine markers express low YAP1 levels [58].

A recent study by Ireland et al. provided evidence that, through the analysis of human and mouse models with single-cell transcriptome analysis, that MYC drives dynamic evolution of SCLC subtypes. In neuroendocrine cells, MYC activates NOTCH signaling inducing dedifferentiation of tumor cells, thus promoting a temporal shift in SCLC from ASCL1[+], to NEUROD1[+], to YAP1[+] non-neuroendocrine states. MYC alternatively promote a consistent transcriptional plasticity. PUO2F3[+] tumors derive from the effect of MYC on

an unknown cell of origin that is a non-neuroendocrine cell [59]. Importantly, the analysis of the transcriptome of single SCLC tumor cells showed that each tumor cell is composed of multiple molecular subtypes, thus supporting the view that SCLC tumor cells possess a moving transcriptional profile triggered by a consistent transcriptional plasticity [59]. According to these findings, it was suggested that SCLCs represent a moving and variable in time therapeutic target, with changing vulnerabilities and thus requiring a combinatorial therapeutic targeting [60].

Wooten et al. reported a systems approach to explore SCLC heterogeneity that integrates transcriptional, mutational, and drug-response data; using this approach, they have identified four SCLC subtypes: Neuroendocrine, non-Neuroendocrine, NEv1 and NEv2 [60]. NE subtype corresponds to SCLC-A, NEv1 to SCLC-N and non-NE to SCLC-Y; NEv2 corresponds to a subtype that has not been described previously and is very similar to SCLC-A subtype and could be considered a SCLC-A2 subtype; this subtype is more resistant than the other subtypes to various drugs [60]. Thus, it was proposed that the SCLC-A subtype comprises two distinct subtypes (SCLC-A and SCLC-A2), with SCLC-A2 being distinguished from SCLC-A by its expression of other factors, such as HES1 [60]. Interestingly, this study also provided evidence that a large proportion of SCLCs comprise more than one subtype; some transcription factors expressed in these tumors act as "master destabilizers" in that their activation destabilizes a phenotype [60].

The main cellular targets of the oncogenic transformation observed in SCLC are represented by rare neuroendocrine stem cells. Pulmonary neuroendocrine cells are neurosensory cells present in the bronchial epithelium promoting epithelium repair following tissue injury. Neuroendocrine stem cells possess a considerable cell plasticity and following an injury dedifferentiate first and are reprogrammed to other cells fates, contributing to tissue repair [60]. Neuroendocrine stem cells are regulated by the genes whose mutations are directly involved in SCLC development: *RB* and *TP53* inhibit self-renewal of neuroendocrine stem cells, whereas *NOTCH* (*NOTCH2*) marks neuroendocrine stem cells and initiates the process of deprogramming and transit amplification [61].

### 4. DNA Methylation Alterations in SCLC

Epigenetic alterations play a key role in the development and maintenance of many tumors, including SCLCs as discussed above, genes encoding various epigenetic factors, such as *KMT2A*, *KMT2B*, *EP300* and *CREBBP* are frequently mutated in SCLC. DNA methylation studies of SCLCs have shown the existence of consistent alterations. A first genome-scale analysis of methylation showed the existence of hundreds tumor-specific methylated genes; particularly, 73 genes were identified that were methylated in most primary SCLC tumors [62]. Interestingly, this analysis showed an enrichment in genes encoding transcription factors and, particularly, of neural cell fate-determining transcription factors, such as NEUROD1, HAND1, ZNF423 and REST [62].

Poirier and coworkers have performed another genome-scale analysis of DNA methylation and defined distinct disease subtypes, which in part correlated with subsets defined by gene expression analysis [63]. Thus, DNA methylation profiling identified three methylation clusters, defined as SCLC M1, M2 and SQ-P: DNA methylation was significantly increased in SCLC M1 and M'' subsets compared to SQ-P; the number of methylated gene promoters was lower in SQ-P, compared to other methylation subtypes [63]. Interestingly, DNA promoter methylation levels in SCLCs are among the most elevated in various tumors reported in the TCGA datasets [63]. *TP53* and *RB1* gene alterations were equally present in the different methylation subtypes; at the level of transcription factor expression, SCLC-M1 tumors are ASCL1$^{low}$/NEUROD1$^{high}$ and SCLC-M2 tumors ASCL1$^{high}$/NEUROD1$^{low}$ [63]. Importantly, this study also showed the expression of the histone methyltransferase EZH2 is particularly high in SCLCs; among the tumors reported in the TCGA and Cancer Cell Line Encyclopedia (CCLE) the EZH2 values observed in SCLC are the highest [63].

Some relevant genes are epigenetically silenced in SCLCs. Thus, the death receptors TRAIL-R1 and TRAIL-R2, caspase-8 and Fas are silenced in SCLC by promoter methylation;

this gene silencing is responsible for the resistance of SCLCs to apoptosis mediated by death receptors [64]. This resistance can be reduced by treatment of SCLC cells with demethylating agents and interferon-γ [64].

## 5. Intratumoral Heterogeneity

The molecular studies of characterization of genomic alterations of SCLC, as well as of other solid tumors, were based on the analysis of a single tumor specimen that is expected to be representative of the whole tumor. However, it is well known that most of solid tumors display a variable and often consistent degree of intratumoral heterogeneity (ITH). It is fundamental to define the degree of ITH of a tumor for its role in tumor evolution and chemoresistance.

Few studies have investigated ITH in SCLC. Zhang et al. explored ITH by multiregional analysis of 34 operative SCLC tumor specimens obtained before systemic therapy; 102 multiregional tumor tissues were obtained from these 34 tumors and analyzed by whole exome sequencing for clonal and subclonal somatic mutations and copy number variants [65]. The most frequently mutated genes in these patients were *TP53* (88%), *RB1* (70%), *TTN* (68%), *TBEB3CL* (65%) and *MUC16* (56%), all with a clonal distribution [66]. These tumors displayed a consistent degree of ITH that was estimated to be 0.50 for mutational ITH and 0.49 for CNV ITH; univariate analysis showed that higher CNV ITH was significant positive predictor of OS, while higher mutational ITH was not associated with OS significantly [65]. Zhou et al. have analyzed by multiregion exome sequencing 120 samples derived from 40 stage I-III SCLC surgically resected [66]. The most frequent mutatnt genes in these tumors were *TP53* (88%), *RB1* (72%) with clonal distribution and *LPR1B* (22%), *PCLO* (15%) and *KMT2D* (15%) with subclonal distribution; a consistent leve of ITH was observed both at mutational (median 0.30) and at CNV levels (median 0.49) [66]. A higher mutational ITH was associated with worse overall survival, whereas a higher tumor burden per cluster was associated with better disease-free survival [66].

To investigate the contribution of ITH to chemoresistance, Stewart et al. have developed models from SCLC patients based on circulating tumor cell (CTC)-derived xenografts and have studied the tumor cells generated in these models by single-cell RNA sequencing [67]. This study was carried out longitudinally using both the CTCs from chemosensitive and chemoresistant SCLC patients [67]. The results of this analysis showed that globally the level of ITH increased with development of chemoresistance, thus suggesting that coexistence of transcriptionally heterogeneous tumor cells with a different spectrum of chemosesitivity/chemoresistance could represent an important mechanism for the evolution of SCLC from chemosensitivity to chemoresistance [67].

## 6. Targeted Therapy and Immunotherapy for SCLC

Most of SCLCs harbor functional inactivation of *TP53* and in part also of *RB1* genes; the targeting of altered *TP53* and *RB1* genes remains now elusive. As above discussed, recent studies using high throughput technologies have shown that SCLCs display a consistent number of additional genetic alterations, including *PTEN* loss, *FGFR1* amplifications, activating *PI3K* mutations, DNA damage response, cell-cycle gene alterations (*ATR*, *CHK1* and *WEE1*), NOTCH pathway, apoptotic pathway, mitotic pathway (Aurora Kinase A) and epigenetic regulation (BET proteins).

Thus, the better understanding of the molecular alterations occurring in SCLC has provided evidence that some of these genetic alterations can be targeted at therapeutic level. The main therapeutic targets and the corresponding drugs and clinical studies are reported in Table 2.

**Table 2.** Treatment targets, predictive biomarker, corresponding drugs and clinical studies.

| Pathway | Target | Clinical Context and Biomarker | Drug and Clinical Studies |
|---|---|---|---|
| DNA Damage Repair (DDR) | PARP1 | Relapsed SLCL. SLFN11 is a biomarker correlated with response. Low inflammatory signature is a biomarker of resistance. | Veripalib or Oripalib with Temozolomide: Improved PFS in SLFN11-positive patients |
| DNA Damage Repair (DDR) | PARP1 | First-line untreated SCLC | Veripalib with standard chemotherapy (platinum + etoposide): improved PFS but not OS. |
| DNA Damage Repair (DDR) | ATR | Relapsed SCLC. CNAs in genes driving replication stress (CCNE1 gain, ARID1A loss) in responding patients | Berzosertib (ATR inhibitor) with Topotecan 1: 36% of responding patients. |
| Cell Cycle | CHK1 | Relapsed SCLC. CHK1 inhibitors synergize with cisplatin. PARP inhibitors and ICIs. CHK1 and MYC overexpression are biomarkers of response to CHK1 inhibitors | Prexaserib, a CHK1 inhibitor alone showed very limited antitumor activity in relapsed SCLC patients. |
| Cell Cycle | WEE1 | Relapsed SCLC | AZD17765, a WEE1 inhibitor, used in monotherapy showed very limited antitumor activity in relapsed SCLC patients. |
| Cell Cycle | CDK4 CDK6 | CDK4/CDK6 inhibitors cannot decrease the proliferations of SCLCs that are RB1-defective, but can decrease the cytotoxic effects of chemotherapy on the hematopoietic system reducing the cycling activity of HSCs and stimulate the anti-tumor immune response | Trilaciclib, a CDK4/CDK6 inhibitor when associated with various chemotherapy regimens in first- or second-line improved chemotherapy-induced myelosuppression but did not modify anti-tumor efficacy outcomes. |
| NOTCH Pathway | DLL3 | Maintenance therapy or second-line therapy. DLL3 tumor positivity is a biomarker of sensitivity. | Rovalpituzumab tesirine as maintenance therapy or as second-line therapy: no effect on OS. |
| NOTCH Pathway | NOTCH2 NOTCH3 | First-line therapy in association with standard chemotherapy (platinum plus etoposide) | Tarextumab in association with platinum plus etoposide in first-line, compared to placebo plus chemotherapy. |
| Apoptosis | BCL-2 | First-line therapy in association with chemotherapy; second-line therapy, relapsed patients. High BCL-2 level is a biomarker of in vitro sensitivity to Venetoclax | Nivotoclax in monotherapy, in relapsed SCLC patients: very low responsding rate. |
| Mitotic pathway | AURORA A KINASE | Relapsed/refractory SCLC in association with chemotherapy c-MYC expression is a biomarker of sensitivity | Alisertib plus plataxel in second line therapy: no benefit in the overall population; improvement of PFS in patients with c-MYC expression. |
| Epigenetic Regulation | BET Proteins | Second-line therapy in association with PARPis or Ventoclax. | No clinical response using BET inhibitors in monotherapy in relapsed/refractory patients. |

### 6.1. Targeting DNA Damage Response (DDR)

Most SCLC patients have a metastatic disease at diagnosis and only 20–30% of patients have an earlier-stage disease, amenable to a chemotherapy plus radiotherapy treatment with a potentially curative intent. The median survival for these patients is low: 9–10 months for metastatic disease and 2 years for non-metastatic patients. The treatment for patients with locally advanced or metastatic disease consists in radiotherapy with concomitant chemotherapy; the first-line chemotherapy for newly diagnoses SCLC consists of a platinum agent (cisplatin or carboplatin) in association with the topoisomerase II inhibitor etoposide. Most of the SCLC patients have disease relapse after a few months of frontline treatment; the treatment of relapsing SCLC patients is very challenging and

there are only two drugs approved by the FDA, topotecan and labinectedin, both exhibiting only modest efficacy. The most accurate predictor of second-line efficacy is the duration of clinical benefit deriving from the first-line therapy with platinum and etoposide.

Recent studies have explored the molecular mechanisms responsible for the rapid development of chemoresistance observed in SCLC after an initial, transient response to first-line chemotherapy. Gardner et al. have explored the mechanisms of acquired resistance to first-line cisplatin and etoposide therapy in SCLC by an experimental approach attempting to mimic the clinical treatment using a set of chemosensitive patient-derived xenograft (PDX) models [68]. The development of chemoresistance in these models was associated with the suppression of the expression of SLFN11 (Schlofen 11, DNA/RNA helicase involved in DNA-damage repair, by blocking replication at stressed replication forks in the presence of DNA damage); SLFN11 silencing was associated with marked increase of H3K27me3, a histone modification promoted by EZH2 [68]. In line with these findings, the treatment with an EZH2 inhibitor restores SLFN11 expression and chemosensitivity; the association of an EZH2 inhibitor with standard chemotherapy prevented emergence of acquired chemoresistance and increased chemotherapeutic efficacy in both chemoresistant and chemosensitive SCLC cells [68]. The same PDX-based approach was used to show that SLFN11 is a biomarker of response of SCLC cells to PARP inhibitors, such as Olaparib, Recuparib and Veliparib; particularly, SCLCs expressing SLFN11 are particularly sensitive to the combinatorial therapy with a PARP inhibitor (talazoparib) and temozolomide, a promising drug combination for the treatment of SCLC patients [69]. In line with these observations, recent clinical studies have shown that SLFN11 expression is a predictive biomarker for response to PARP inhibitors. SLFN11 expression by immunohistochemistry was correlated with outcomes in the randomized phase II trial of temozolomide (a DNA damaging agent) plus veliparib (a PARP inhibitor) versus temozolomide plus placebo in a group of SCLC patients with relapsed disease: in the whole population of patients there were no significant differences in both PFS and OS between the two arms of treatment; in patients receiving temozolomide plus velaparib, outcomes were superior in patients SLFN11-positive compared to those SLFN11-negative (progression-free survival: 5.7 vs. 3.6 months; overall survival: 12.2 vs. 7.5 months); in patients treated with temozolomide plus placebo there was no difference in response in patients SLFN11-negative and SLFN11-positive [70]. In another phase I/II study, the PARP inhibitor Olaparib in combination with TMZ was evaluated in a cohort of relapsed SCLC patients reporting an ORR of 41%, a PFS of 4.2 months and a median OS of 8.5 months [70]. Biomarker investigation showed that a low basal expression of inflammatory response genes correlated with resistance to Olaparib plus TMZ treatment [71]. These observations have supported the first clinical trial in which SLFN11 was used as a potential biomarker predictive for response to PARPis. Thus, the SWOG 1929 trial is evaluating maintenance of atezolizumab plus talazoparib versus atezolizumab alone in advanced stage SCLC; in this study, all patients were initially treated with frontline therapy based on EP plus atezolizumab and were concomitantly screened for SLFN11 positivity by immunohistochemistry: SLFN11-positive patients are then randomized in the maintenance phase to one of the two above mentioned treatments.

In a very recent study, Willis et al. performed a retrospective analysis on 124 SCLC patients explored for tumor SLFN11 expression by immunohistochemistry and observed that about 27% of these tumors display low/absent SLFN11 expression; furthermore, in 18% of these tumors SLFN11 expression was subclonal [72]. Interestingly, a retrospective analysis on the response of these patients, subdivided into two groups (low and high) according to SLFN11 expression levels, to standard EP chemotherapy treatment showed that SLFN11 high patients had significantly improved PFS and OS compared to SLFN11 low patients [72]. This difference was observed also in patients treated in first line with any kind of therapy [72]. Qu et al. have analyzed the molecular classification of 146 primary SCLC tumors and SLFN11 expression by immunohistochemistry: SLFN11 expression was absent in 40% of SCLCs, particularly in those negative for subtype markers [73].

Since SCLC tumor tissue availability is limited for investigation SLFN11 expression, Zhang et al. have recently shown that SLFN11 expression can be detected in circulating tumor cells: at diagnosis, about 50% of patients exhibited SLN11 expression in tumor cells, but this positivity decreased to 25% in patients receiving platinum treatment [74]. SLFN11 expression in positive cases was heterogeneous at the interpatient and intra-patient levels, independent of tumor morphologic subtype.

Recent studies have further explored the role of methylation mechanisms in the inhibition of SLFN11 expression in SCLC. Thus, Krushkal et al. showed that increased promoter methylation of *SLFN11* was correlated with resistance to DNA damaging agents, because of low or no *SLFN11* expression [75]. SLFN11 is involved also in the mechanisms of resistance to pyrrolobezodiazepine (PBD)-conjugated antibody-drug conjugates (ADC): *SLFN11* silencing reduced sensitivity to PBD-ADC and treatment with an EZH2 inhibitor derepressed SLFN11 expression in PBD-ADC-resistant cell lines and increased their sensitivity to these drugs [76]. These preclinical data have supported an ongoing phase I/II clinical trial (NCT03879798) exploring the clinical efficacy of the drug combination DS-3201b (EZH2 1–2 inhibitor) and irinotecan in recurrent SCLC.

As above mentioned, the only second-line approved agents for SCLC treatment are Topotecan an Lurbinectedin. Lurbinectedin is a synthetic analog of the natural marine-based tetrahydroisoquinoline, trabectedin which is derived from the sea squirt species *Ecteinascidia turbinate*. In a phase II basket trial Lubinectedin was shown to induce a significant response in 35% of patients in second-line treatment [77]. Studies on SCLC cell liens have shown that SLFN11 positivity is a marker of sensitivity to Lurbinectedin; the low sensitivity to Lubinectedin displayed by SCLC cell lines exhibiting low SLFN11 expression was restored through the addition of an ATR inhibitor [78].

Aberrant expression of DNA damage repair genes in SCLC has been reported [65]. Among these genes, *PARP1* was highly expressed at mRNA and protein level in SCLC, which together with Burkitt lymphoma, is the tumor more expressing PARP1; at the level of lung cancers, SCLC is the tumor most expressing PARP1 [79]. Mutations of DNA repair pathways are enriched in post-treatment samples [43]. Park et al. reported the results of targeted gene sequencing of 166 SCLC patients with extensive or limited disease and explored the clinicopathological implications of DDR pathway alterations in these patients, including DNA double-strand breaks repair pathway genes and single-strand breaks repair genes; 42% patients displayed intact DDR genes and 58% altered DDR genes [80].

Other studies have supported the use of PARP inhibitors in SCLC patients. These studies were promoted by a proteomic preclinical study that led to the identification of PARP inhibitors and other DNA damage response inhibitors as potential drugs to target a vulnerability of SCLC [79]. This observation has promoted several clinical studies aiming to evaluate the capacity of a PARP inhibitor Veliparib, to improve the clinical benefit deriving from the standard chemotherapy with platinum and etoposide. These studies have shown that Veliparib plus platinum chemotherapy was able to improve PFS as first-line treatment for SCLC; however, this improvement in PFS was not associated with a concomitant benefit in OS [79,81–83]. In one of these studies, SLFN11 positivity showed a trend toward improved response in the group treated with Veliparib compared to control [82].

Whole-exome sequencing studies have shown that a minority of SCLC patients displayed germline deleterious variants in some cancer-predisposing genes; particularly, SCLC patients were more likely to carry germline *RAD51D*, *CHEK1*, *BRCA2* and *MTYH* pathogenic variants than healthy controls [84]. Interestingly, treatment of a patient with relapsed SCLC and a germline mutation of *BRIP1*, a homologous-recombination-related gene, using PARP inhibitors resulted in a remarkable disease response [84].

The clinical trials of PARP inhibitors in combination with chemotherapy have shown only limited responses and future ongoing studies are required to identify biomarkers predicting response to PARP inhibitors and drug combinations able to improve clinical benefit in terms of survival. The combination of PARP inhibitors with agents that damage

DNA and inhibit DNA damage response appears to be particularly effective in preclinical studies and in initial phase I clinical studies [85]. Expression of SLFN11 and other molecules involved in the DNA damage response seems to select for improved clinical responses [85].

Several ongoing clinical trials involving various combinations of PARP inhibitors (such as Oleparib, Talazoparib, Rucaparib and Niraparib) with chemotherapy, immunotherapy and VEGF inhibitors are reported in Table 3.

**Table 3.** Ongoing clinical trials in SCLC patients involving PARP inhibitors.

| Trial Number | Study Population | Trial Design | Medication | Phase | Peculiar Features of the Trial |
|---|---|---|---|---|---|
| NCT02511795 | Relapsed/refractory ES-SCLC | Phase Ib including a dose-escalation and a dose-expansion phases | AZD1775 (WEE1 TK inhibitor) + Olaparib | Ib | |
| NCT03532880 | ES-SCLC | Phase I including patients with E-SCLC who have completed induction chemotherapy and have no evidence of tumor growth | Olaparib + low-dose radiotherapy | I | Patients with ES-SCLC with stable disease after chemotherapy |
| NCT04939662 | Relapsed/refractory ES-SCLC | Single-arm phase II study of Olaparib and Bevacizumab combination therapy in SCLC as second/third line therapy. Enrolled patients must display ATM deficiency, SLFN11 positivity, POU2F3+; HR pathway gene mutation | Olaparib + Bevacizumab (VEGF inhibitor) | II | Samples from primary and metastatic tumors in progression are investigated for analysis of the mechanisms of resistance |
| NCT02498613 | Relapsed/refractory ES-SCLC | Single-arm phase II study of Olaparib and Cediranib combination in metastatic SCLC patients who received one prior line of platinum chemotherapy | Olaparib + Cediranib (VEGF inhibitor) | II | Correlation with DNA repair gene expression. Promising clinical activity in biomarker unselected patients with ORR of 28%. |
| NCT02769962 | Relapsed/refractory ES-SCLC | Single-arm phase I/II study of Olaparib with CLRX101 | Olaparib + CLRX101 (nanoparticle Campothecin) | I | |
| NCT02484404 | Patients previously treated with chemotherapy | Single-arm phase I/II study of Olaparib with MEDI4736, an anti-PD-L1 Ab | Olaparib + MEDI4736 (PD-L1 Ab) | I/II | |
| NCT02734004 | Relapsed/refractory ES-SCLC | Single-arm phase I/II study of Olaparib with MEDI4736, an anti-PD-L1 Ab | Olaparib + MEDI4736 (PD-L1 Ab) | I/II | |
| NCT04538378 | Lung cancers with EGFR mutations developing resistance to EGFR inhibitors through transformation to SCLC | Single-arm phase II study of Olaparib with Durvalumab, an anti-PD-L1 Ab | Olaparib + Durvalumab (PD-L1 Ab) | II | |
| NCT04170946 | Patients with ES-SCLC with stable disease after standard induction chemotherapy | Single-arm phase I study of Talazoparib in combination with consolidative thoracic radiotherapy | Talazoparib + Consolidative radiotherapy | I | Maintenance therapy for stable disease after induction chemotherapy |
| NCT04334941 | Patients with SLFN11 positive ES-SCLC | Randomized phase II study of maintenance versus Atezolizumab with Talazoparib | Talazoparib + Atezolizumab | II | |
| NCT03672773 | Patients with ES-SCLC with stable disease after standard induction chemotherapy | Single-arm phase II study of Talozapir in combination with Temozolomide | Talazoparib + Temozolomide | II | Intermittent low-dose Temozolomide in association with continuous talazoparib |

**Table 3.** *Cont.*

| Trial Number | Study Population | Trial Design | Medication | Phase | Peculiar Features of the Trial |
|---|---|---|---|---|---|
| NCT03958045 | ES-SCLC | Single-arm phase II study of Rucapirib in combination with Nivolumab | Rucaparib + Nivolumab (PD-1 Ab) | II | Maintenance therapy for stable disease after induction chemotherapy |
| NCT04209595 | ES-SCLC | Single-arm phase I/II study of Rucapirib in combination with PLX038 | Rucaparib + PLX038 (PEGylated conjugate of irinotecan) | I/II | Improvement in DNA damage induced by PLX038 |
| NCT03830918 | Patients with ES-SCLC with stable disease after standard induction chemotherapy | Niraparib + Temozolomide + Atezolizumab vs Atezolizumab as maintenance therapy in ES-SCLC | Niraparib + Atezolizumab + Temozolomide | | Maintenance therapy for stable disease after induction chemotherapy |
| NCT04701307 | Patients with ES-SCLC with stable disease after standard induction chemotherapy | Single-arm phase II study of Nirapirib in combination with Dosratlimab | Niraparib + Dostarlimab (anti-PDCD1 Ab) | II | |

A recent study by Thomas et al. showed that SCLCs are particularly sensitive to drugs targeting the replication stress response [86]. This vulnerability is dependent on the genetic alterations occurring in SCLCs predisposing to acceleration of S-phase entry and disruption of the DNA replication schedule, abnormalities characterizing a condition of replication stress [72]. These authors have explored in vitro drug response profiles of SCLC and observed that pharmacological inhibition of ataxia telangiectasia mutated and RAD3 related (ATR), the primary target of stress replication response increased cell death and improved anti-tumor activity of DNA topoisomerase I [86]. The demonstration of enhanced efficacy of an ATR inhibitor with topotecan, the most used drug for SCLC therapy in second-line, provided a strong rationale for a combination therapy clinical trial. Thus, in a proof-of-concept phase II clinical trial the ATR inhibitor berzosertib (M6620) was administered together with topotecan to SCLC patients who have relapsed after at least one prior chemotherapy, resulting in an overall response rate of 36%, with 30% of responses among patients with platinum-resistant disease [86]. After a follow-up of 20.7 months, the PFS at 4 and 6 months was 60% and 36% and the median overall survival was 8.5 months (at 6- and 12-months OS was of 68% and 32%, respectively). The median duration of response was 6.4 months [86]. Some genomic alterations correlated with response to ATR and TOP1 inhibition: (i) tumor mutation burden and specific gene mutations were not significantly associated with response to treatment; (ii) 50% of the responding tumors displayed copy number alterations in genes driving replication stress, such as *CCNE1* gain and *ARID1A* loss; 33% of the responding patients (corresponding to cases with no copy number alterations at the level of replication stress-related genes) exhibited focal gains in *SOX2* and *SOX4* and correspond to platinum-resistant patients [86]. The analysis of transcriptomes at the level of responding and non-responding patients showed some remarkable differences. Responding tumors exhibited an enrichment for gene pathways involved in cell-cycle progression and DNA repair, an increased expression of genes upregulated in response to activation of the ATR pathway and downregulation of genes involved in immune response, metabolism and adhesion. Genes involved in neuronal and neuroendocrine fate are markedly enriched among responding tumors (about 44% of tumors with neuroendocrine phenotype responded to treatment, compared to 0% of those with low neuroendocrine phenotype). All the responding tumors corresponded to NE ASCL1 or NEUROD1 subtypes, whereas no tumors corresponding to the non-NE POUF3 and YAP1 subtypes responded to treatment with ATR and TOP1 inhibition [86]. The findings of this study need to be confirmed in a phase III clinical trial in a larger cohort of relapsed SCLC.

In addition to these findings, Nagel et al. confirmed the vulnerability of SCLCs to ATR inhibitors and also showed a consistent synergism between ATR inhibitors and

cisplatin in in vivo mouse SCLC models, outperforming the combination of cisplatin with etoposide [87]. These observations strongly support the study of the drug combination, ATR inhibitors and cisplatin, in clinical studies [87].

Several ongoing clinical trials are evaluating various combinations of ATR inhibitors (Berzosertib and Cerlosertib) with chemotherapy, immunotherapy or PARP inhibitors (Table 4).

**Table 4.** Ongoing clinical trials involving the combination of ATR inhibitors with chemotherapy, immunotherapy or PARP inhibitors.

| Trial Number | Study Population | Trial Design | Medication | Phase |
|---|---|---|---|---|
| NCT02157792 | ES-SCLC | Phase I study involving treatment with the ATR inhibitor Berzosertib and chemotehrapy | Berzosertib (VX-970 M6620) + Cisplatin/ Etoposide or Cisplatin | I |
| NCT02487095 | Refractory/relapsed ES-SCLC | Phase I/II study involving treatment with ATR inhibitor Berzosertib and Topotecan | Berzosertib + Topotecan | I/II |
| NCT04768296 | Refractory/relapsed platinum-resistant ES-SCLC | Single-arm phase II study involving treatment with ATR inhibitor Berzosertib and Topotecan | Berzosertib + Topotecan | II |
| NCT03896503 | Refractory/relapsed ES-SCLC | Randomized trial involing treatment with Topotecan alone or in combination with Berzosertib | Berzosertib + Topotecan | II |
| NCT0328607 | Refractory/relapsed ES-SCLC | Two-arms phase II study involving the treatment of refractory/relapsed ES-SCLC with either Olaparib alone (patients with HR pathway gene mutations) or in association with Ceralasertib (unselected patients) | Olaparib + Ceralasertib (AZD6738) | II |
| NTC04361825 | Refractory/relapsed ES-SCLC | Single-arm phase II study involving treatment with ATR inhibitor Cerlasertib and the anti anti-PD-L1 Ab Durvalumab | Cerlasertib + Durvalumab | II |

A recent study based on RNA single-cell sequencing studies provided evidence about a new molecular mechanism of platinum resistance based on a transcription shift. Using ASCL1-positive CDX models derived from treatment-naïve patients and studied these cells during the status of platinum-sensitivity and platinum-resistance: in these cells, platinum resistance was accompanied by a shift from ASCL1-positivity to ASCL1-negativity; importantly, SCXL1-negative cells, defined as SCLC-I cells, did not gain NEUROF1, POUF3 or YAP1 expression [56]. The marked phenotypic changes were associated with marked fluctuations in NOTCH pathway activation and acquisition of stem like features, such as differentiation plasticity, that may contribute to the capacity of these cells to mediate platinum resistance [56].

Other recent studies have defined other mechanisms of chemoresistance in SCLC. A recent study by Jin et al. has shown that activation of PI3K/AKT pathway is a potential mechanism of chemoresistance in SCLC [88]. Twenty L > S-SCLC patients at baseline and at relapse were explored; targeted NGS showed that at relapse genes in the PI3K/AKT signaling pathway were enriched for acquired somatic mutations or high frequency of acquired CNVs [88]. Pathway analysis on differentially upregulated proteins in relapsed SCLC showed activation of HIF-1 pathway [88]. Targeting of PI3K/AKT pathway may represent one potential mechanism to bypass therapeutic resistance of SCLC [88].

Quintal-Villelonga et al. have evaluated the drug sensitivity of SCLC cell lines representing different molecular subtypes. This screening identified exportin-1, encoded by *XPO1*, as a therapeutic target: the small molecule exportin-1 inhibitor selinexor in combination with cisplatin or ironetac markedly inhibited tumor growth in chemonaive and chemorelapsed SCLC PDXs, respectively [89].

Schenk et al. have exploited paired pre-treatment and post-chemotherapy CTC PDX models from SCLC patients to explore changes occurring at disease progression after chemotherapy [90]. Using this approach, they showed that soluble guanylate cyclase (sGC) is commonly upregulated in post-chemotherapy progression PDX models; expression and activation of sGC is regulated by NOTCH and nitric oxide (NO) signaling; inhibition of sGC or pharmacological targeting of NO synthase elicited a resensitization of PDX-resistant models to chemotherapy [90]. These observations suggest a new approach to bypass chemoresistance in SCLC.

*6.2. Cell-Cycle Targeting*

Two experimental studies supported a possible therapeutic targeting of Checkpoint Kinase 1 (CHK1) in SCLC. SCLCs display a higher expression of CHK1 protein than NSCLCs. Sen et al. have explored the consequences of *TP53* and *RB1* loss, very frequently observed in SCLC, resulting in an increased expression of E2F1 for lack of repression and increased levels of several proteins involved in DNA damage repair, such as PARP1 and CHK1 [91]. The function of CHK1 is peculiar in that this protein, in cells where the *TP53* function is deregulated by mutations or deletion events, becomes a key mediator of cell cycle arrest induced by DNA damage [91]. Using various SCLC models, it was provided evidence about a marked sensitivity of SCLC cells to CHK1 inhibition by RNA interference or to CHK1 inhibitors; importantly, the treatment of SCLC cells with a CHK1 inhibitor, such as prexasertib potentiated the antitumor activity in vitro and in vivo of cisplatin of the PARP inhibitor Olaparib [91]. Proteomic analysis suggested that CHK1 and MYC are the main predictive biomarkers of LY2606368 sensitivity, thus suggesting a potential efficacy of CHK1 inhibition in SCLC with MYC amplification/overexpression [90].

Doerr et al. have made a transcriptomic comparative study of lung cancers, including both SCLC and NSCLC, providing evidence about a significantly increased expression of DNA damage response in SCLC, compared to NSCLC [92]. Particularly, this analysis showed that the genes encoding for *CKH1*, *CDC25A*, *CDC25B* and *CDC25C* are overexpressed in SCLC compared to lung adenocarcinoma. Importantly, both CHK1 and ATR inhibitors induced apoptosis in SCLC cells but not in lung adenocarcinoma cells [92]. These observations have supported the ATR/CKH1 cell cycle axis as a possible therapeutic target for the treatment of SCLC.

More recent studies have shown the antitumor synergism of CHK1 inhibitors with other anti-tumor drugs. Thus, Hsu et al. provided evidence that CHK1 inhibition synergizes with cisplatin in inducing mitotic cell death in *TP53*-deficient SCLC cells. Furthermore, in in vitro and in vivo mouse models of SCLC resistant to platinum compounds, CHK1 inhibitors overcame cisplatin resistance; this observation supports a combinatorial strategy based on a CHK1 inhibitor and cisplatin for SCLC treatment [93]. Other studies have shown the capacity of CHK1 inhibitors to potentiate the immune anti-tumor mechanisms regulated by immune checkpoints. Thus, Sen et al. provided evidence that targeting of SCLC tumor cells with either CHK1 or PARP inhibitors consistently enhanced the surface expression of the protein PD-L1 and this mechanism potentiated the anti-tumor effect of PD-L1 blockade and increase tumor infiltration by cytotoxic T-lymphocytes in various SCLC in vivo models [94]. Furthermore, both CHK1 and PARP inhibitors activated the STING innate immune response pathway, resulting in consequent activation of the cytotoxic activity of T-lymphocytes [95]. These observations provide an important rationale for combining CHK1 or PARP inhibitors with immune checkpoint inhibitors to enhance treatment efficacy of immunotherapy in patients with SCLC [96]. The same authors showed also that a combination treatment with a CHK1 inhibitor and low-dose gemcitabine increases the anti-tumor effect of PD-L1 blockade via a modulation of the immune microenvironment in SCLC [97].

A recent phase II study assessed the safety and efficacy of Prexasertib, a CHK1 inhibitor, in SCLC patients who progressed after standard therapies and showed platinum-sensitive or platinum-resistant disease: 5.2% of platinum-sensitive and 0% of platinum-resistant

SCLCs showed a response to the treatment [96]. These results suggest that Prexasertib used alone in monotherapy did not demonstrate significant anti-tumor activity in relapsing SCLCs.

The expression of the GM2M checkpoint regulator WEE1 is increased in SCLCs compared to normal lung tissue; this activity is relevant for the survival of SCLC cells, as shown by their sensitivity to the WEE1 inhibitor AZD1775 [98]. In preliminary phase I studies, the AZD17775 inhibitor showed some activity in a subset of SCLC patients, but resistance rapidly develops. Resistance to WEEI inhibitors is promoted by AXL through downstream mTOR signaling, inducing the activation of a parallel DNA damage repair pathway, mediated by CHK1 [97]. A phase II umbrella trial failed to show a significant anti-tumor activity of AZD17765, when evaluated in monotherapy in SCLC patients who have failed prior platinum-based chemotherapy [98].

Preclinical studies have shown a marked anti-tumor synergism of the WEEI inhibitor AZD1775 and the PARP inhibitor Olaparib, resulting in a more pronounced inhibitory effect on chemosensitive SCLC tumor explants than platinum plus etoposide [99].

A very recent study also explored the mechanisms of acquired resistance of SCLC cells to CHK1 inhibitors that seem to be mediated by overexpression of the cell cycle regulator WEE1 [100].

*6.3. NOTCH Pathway Targeting*

NOTCH ligands DLL1, DLL4, JAG1 and JAG2 activate NOTCH receptor signaling; in contrast, DLL3 is localized at the level of the Golgi apparatus and is unable to activate NOTCH signaling. Classic SCLCs display high expression of DLL3, while lower levels are observed in variant SCLC [101]. Importantly, DLL3 protein was expressed on the surface of tumor cells but not in normal adult tissues [101]. Elevated expression of DLL3 was observed only at the level of high-grade tumors (SCLC and LCNEC): 47% and 75% of nSCLCs and 49% and 54% of LCNECs displayed a high DLL3 expression, according to H-score (percentage of positive cells by staining intensity) and percentage of tumor positive cells, respectively [102].

A DLL3-targeted antibody-drug conjugate, SC16LD6.5, composed of a humanized anti-DLL3 monoclonal antibody conjugated to a DNA-damaging pyrrolobenzodiazepine (PDD) dimer toxin was developed. This drug conjugate was able to induce durable tumor regression in SCLC PDX models, with an intensity related to the level of DLL3 expression on tumor cells. These observations have suggested that DLL3-targeting could represent a promising strategy for the treatment of high-grade pulmonary neuroendocrine tumors [101]. Other studies have confirmed the frequent and elevated expression of DLL3 in SCLC. At the immunohistochemical level, 83% of SCLCs display DLL3 expression, with 32% of cases showing >50% of DLL3-positive cells (DLL3-high); the survival of DLL3-high and DLL3-low SCLCs was similar [103]. A recent large study on 1073 SCLC specimens showed DLL3 expression in 85% of cases, with 68% of high positivity [104]. DLL3 expression had no prognostic impact. Baine et al. have explored by immunohistochemistry DLL3 expression in various SCLC subtypes and showed that DLL3 was highly expressed in ASCL1-dominant and NEUROD1-dominant subtypes, whereas it was entirely negative in ASCL1/NEUROD1 double-negative SCLCs [54]. In line with these findings, Hu et al. showed in a cohort of 247 surgically resected LS-SCLC that DLL3-high expression was observed in 72.8 % of cases and was positively associated with ASCL1 expression, vascular invasion and strongly correlated with the expression of thyroid transcription factor-1 (TTF1) and neuroendocrine markers [105].

The anti-tumor effect of rovalpituzumab tesirine (DLL3-specific IgG1 monoclonal antibody conjugated with a potent DNA cross-linking agent, Rova-T) was evaluated in a phase I study enrolling 82 patients, comprising 74 SCLCs and 8 large-cell neuroendocrine carcinomas. The clinical response to the treatment was evaluated in 60 patients and 18% of them displayed an objective response; 38% of the patients exhibiting DLL3 expression on tumor cells displayed a response to the treatment [106]. Thus, rovalpituzumab tesirine

showed encouraging single-agent antitumor activity with a manageable safety profile in a population of heavily pretreated SCLC patients [106]. From these initial studies, it emerged that an elevated DLL3 expression in tumor cells is an important biomarker in clinical studies based on the use of rovalpituzumab. In phase III clinical trials, the evaluation of the level of DLL3 expression on SCLC cells is a key determinant of the tumor response. To facilitate the evaluation of DLL3 expression on SCLC tumor cells and to select patients for the treatment with Rova-T, Sharma et al. have developed a non-invasive exploration of the in vivo DLL3 expression status using $^{89}$Zr-labeled SC16 anti-DLL3 antibody and PET imaging [107]. The introduction of this new methodology based on the use of radiolabeled anti-DLL3 antibody could represent a precious tool for both selection of patients suitable for therapy with Rova-T and for the evaluation of the response to therapy [107].

TAHOE and MERU were both randomized phase III clinical trials of Rova-T in the second-line and maintenance setting, respectively; however, both these two studies were terminated early for failure to meet interim progression-free survival and overall survival endpoints [108]. Thus, the phase III TAHOE study failed to show any efficacy of Rova-T compared with Topotecan as second-line therapy in DLL3-high SCLC [109]. The MERU study failed to show any advantage in overall survival using Rova-T as a maintenance therapy after platinum-chemotherapy in patients with extensive-stage SCLC [110]. Furthermore, the results of the phase II TRINITY study carried out in 339 SCLC patients with DLL3-positive tumors, Rova-T showed only modest clinical effects, with a response rate of 12.4% and an overall survival of 5.6 months [111]. The development of Rova-T was stopped.

Other agents able to target DLL3 on the surface of SCLC cells are under development. One of these agents is represented by bispecific T cell engager (BITE) able to target and to crosslink DLL3 on SCLC cells with CD3-positive T lymphocytes, promoting T cell-mediated tumor lysis [112]. Hipp et al. Reported the development of an IG-like T-cell engaging bispecific antibody (ITE) that redirects T-cells to specifically lyse SCLC cells expressing DLL3 (DLL3/CD3 ITE) in vitro in cell lines and in vivo in an SCLC xenograft model reconstituted with human CD3$^+$ T-cells [112]. Giffin et al. reported the preclinical characterization of AMG 757 in SCLC cell lines and xenograft >SCLC models [113]. In vitro studies have shown that AMG 757 displayed potent and specific killing of SCLC cell lines, including also those expressing low/very low DLL3 levels; in in vivo models, AMG 757 was able to engage systematically administered human T lymphocytes, to induce T cell activation, and to redirect T lymphocytes to lyse tumor cells [113]. In nonhuman primates, AMG 757 was well tolerated and exhibited an extended half-life [113]. In the NCT 03319940 clinical trial the safety profile and the clinical efficacy of AMG 757 was evaluated in 64 SCLC patients with extensive disease who progressed after ≥1 platinum-based regimens, using 10 different AMG 757 doses up to 100 mg (every two weeks) [114]. A total of 13% of the treated patients displayed a partial response, with a duration of response >6 months; in the group of patients treated with the highest AMG 757 dose (100 mg), 5/8 patients displayed a partial response [114]. The study is still ongoing, and the results observed at 100 mg of AMG 757 need to be confirmed in an expanded cohort of SCLC patients. Interestingly, this clinical trial will evaluate also the safety profile and the efficacy of AMG 757 when administered with an anti-PD-1 antibody. This trial is supported by a preclinical study showing that the combined administration of a DLL3-targeted bispecific antibody with PD-1 inhibition efficiently suppress SCLC growth in SCLC xenograft mouse models [115].

A recent study reported the study of near infrared photoimmunotherapy using the rovalpituzumab antibody conjugated with the IR700 photosensitizer, showing promising anti-tumor activity in preclinical models [116]. AMG 119 is an autologous T cell that has been genetically manipulated to express a chimeric antigen T cell receptor addressed against DLL3, resulting in lysis of DLL3-positive cells and autologous T cell proliferation.

NOTCH signaling regulates cell fate decisions during normal development in different organs, including the lung. Particularly, NOTCH signaling exerts an inhibitory effect on neuroendocrine cell differentiation, mainly through DLL-mediated receptor activation. The study of 67 SCLC patients at an extensive stage of disease showed that NOTCH1/2/3

mutations observed in 34% of patients were more frequently observed in lung tissue than in metastatic sites and were associated with an increased overall survival and progression-free survival [117]. NOTCH1 high-expression was observed in a limited number of SCLCs (21%) and was negatively associated with ASCL1 expression; NOTCH1-high expression was associated with a shorter overall survival (8.1 months) compared to NOTCH1-low expression (12.4 months) [118]. In a multivariate analysis, NOTCH1-high expression was a negative prognostic factor [118].

In SCLCs loss-of-function mutations of *NOTCH* genes are frequently observed and ectopic NOTCH signaling induces a tumor suppressive effect. A study by Lim and coworkers better defined the role of NOTCH signaling in SCLC [119]. This study was based on the analysis of NOTCH receptors and ligand expression in a conditional triple-knockout mice *TP53*, *RB*, *p130* SCLC mouse model, showing that high NOTCH expressing tumors were less proliferative and tumorigenic compared to low NOTCH expressing tumors. The consequences of NOTCH signaling are double and may lead to a tumor-suppressive effect reducing the growth of neuroendocrine cells and to a pro-tumorigenic effect by promoting the generation of non-neuroendocrine cells that are chemoresistant and exert a trophic effect on neuroendocrine cells. In SCLC cell lines, NOTCH ectopic expression determines first a growth suppressive effect and then an inhibition of both ASCL1 expression and of neuroendocrine gene expression; however, ASCL1 silencing was not sufficient for inhibition of neuroendocrine differentiation, but it is required also the expression of the transcriptional repressor REST [97]. Importantly, NOTCH signaling inhibition in combination with chemotherapy resulted in a potent suppression of tumor growth and delayed tumor relapse [97]. These findings have represented the rational basis for the development of a clinical trial based on the combination of NOTCH inhibition with chemotherapy in SCLC patients with advanced disease and with tumor cells exhibiting active NOTCH signaling [119]. A randomized phase II clinical study evaluated the combination of an anti-NOTCH 2/3 antibody (Tarextumab) with standard chemotherapy compared to placebo with standard chemotherapy in previously untreated patients: tarextumab in combination with platinum-based chemotherapy failed to improve OS, PFS and ORR [120]. Furthermore, biomarker analysis failed to identify a predictive marker for Tarextumab efficacy [120].

Other studies support a key role of NOTCH inhibition in the mechanisms promoting neuroendocrine differentiation in SCLCs. Under physiological conditions, Rb1 binds to and inhibits the activity of the histone demethylase KDM5A; *RB1* inactivation in SCLC determines KDM5A activation, required to sustain ASCL1 levels and neuroendocrine differentiation [21]. The effect of KDM5A promoting neuroendocrine differentiation is mediated not only though stimulation of ASCL1 expression, but also through a repression of *NOTCH2* and *NOTCH2* target genes [21].

Lysine-specific demethylase 1 (LSD1) is an important epigenetic target for cancer therapy. LSD1 is highly expressed in SCLC relative to other lung cancer subtypes [121]. The sensitivity of SCLC cell lines to A LSD1 inhibitor (GSK 690) is highly heterogeneous and this heterogeneity is mainly related to the differentiation status of SCLC in that tumors with neuroendocrine signatures are sensitive, while tumors with mesenchymal signatures are resistant to LSD1 inhibitors [121]. Lysine demethylase inhibitors exert an inhibitory effect on SCLC, mediated through activation of the NOTCH pathway, with consequent inhibition of ASCL1 expression and repression of SCLC tumorigenesis [122]. A recent study reported the preliminary results of the phase Ib clinical trial NCT 03850067 involving the treatment of extensive stage SCLC patients with the LSD1 inhibitor CC-90011 with etoposide and carboplatin; responding patients received a maintenance dose of CC-90011 once/week [123]. Among 24 enrolled patients, 19 treated patients, efficacy-evaluable achieved a partial response [123]. The study is ongoing to further evaluate CC-90011 with etoposide and carboplatin and with nivolumab.

### 6.4. Potential Therapeutic Targets in ASCL1-Driven SCLC: BCL2

ASCL1 is essential for both the proper development of neuroendocrine cells and is essential for the growth and survival of neuroendocrine lung cancers. The definition of the biochemical network through which the transcription factor ASCL1 orchestrates the neuroendocrine differentiation of SCLC-A subtype is of fundamental importance for the identification of potential therapeutic targets. Both NKX2-1 and PROX1 transcription factors are highly co-expressed with ASCL1; in fact, ASCL1 physically interacts with NKX2-1 and PROX1 [124]. These factors bind overlapping genomic regions and co-regulate a set of genes, including the cell surface proteins, SNC3A and KCNB2, abundantly expressed in SCLC-A subtype [124]. Importantly, genetic depletion of NKX2-1 or PROX1 alone, or in combination with ASCL1, do not result in an inhibitory effect on cell growth superior to that elicited by depletion of ASCL1 alone [124]. ASCL1 forms molecular complexes with NKX2-1 and PROX1 and regulates a set of genes involved in NOTCH signaling, NE-specific genes involved in catecholamine biosynthesis and cell-cycle regulation [125]. Studies on SCLC cell lines have shown that ASCL1 expression is highly correlated with expression of the cancer-driving genes *RET*, *SOX1*, *SOX2*, *FOXA1* and *FOXA2*; furthermore, the expression of the NOTCH inhibitor DLL3 is also significantly correlated with ASCL1 [126]. In a clinicopathological study based on the analysis of 247 SCLC cases, 42.5% highly expressed ASCL1: the comparative analysis of SCLCs subdivided according to the level of ASCL1 expression showed that highly expressing tumors have a higher tumor stage, more frequent lymph node metastases and nerve invasion and reduced overall survival compared to those with low ASCL1 expression [127]. The definition of a 72-gene cell signature associated with ASCL1 expression in SCLC allowed to identify several molecular targets, such as BCL-2 that represent molecular vulnerabilities that can be exploited for therapeutic use [128]. In SCLC, the expression of the gene *SCNN1A* is highly correlated with ASCL1 expression; this gene encodes the alpha subunit of the epithelial sodium channel, whose function can be inhibited by the orally potassium-sparing diuretic amiloride [129].

A recent study showed that ASCL1 is a transcriptional activator of Dopamine and cAMP-regulated phosphoprotein, Mr 32000 (DARPP-32) and its N-terminally truncated splice variant (t-DARRP); DARRP isoforms are overexpressed in SCLC, promote tumor growth and could represent a new therapeutic target [130].

As reported above BCL-2 is a potentially important target of ASCL1 in SCLC. In vitro and in vivo studies in SCLC models have shown the capacity of Navitoclax, a BCL-2/BCLX$_L$ inhibitor, to decrease the growth of SCLC cells [108] These observations supported a clinical study designed to evaluate Navitoclax as a single agent against advanced and recurrent SCLC; however, in this clinical setting, Navitoclax showed only a very limited antitumor activity [131]. However, though the analysis of SCLC cell lines and patient-derived xenografts, it was reached the conclusion that Venetoclax, a potent BCL-2 inhibitor, is effective in SCLCs with high BCL-2 expression [132]. This study suggested rationale for biomarker-guided clinical trials of Venetoclax in high BCL-2-expresssing SCLCs [132]. Venetoclax is being to be evaluated in clinical trials in untreated SCLC patients with extensive disease in association with atezolizumab, carboplatin and etoposide and in relapsed SCLC patients in association with irinotecan. In a recent study, Yasuda et al. have explored the pattern of expression of Bcl2 members in SCLC. Tumors expressing high levels of MCL1 and low levels of Bcl-X$_L$ and Bcl2 are the most common type (48%) and mainly account for POUF3$^+$ and transcription factor negative SCLCs; tumors with high MCL1 and Bcl-X$_L$ and low Bcl2 represent 28% of all SCLCs and mainly account for ASCL1$^+$ and NEUROD1$^+$ SCLCs; tumors expressing low levels of MCL1, Bcl-X$_L$ and Bcl2 represent 12% of all SCLCs and mainly account for ASCL1$^+$ and POUF3$^+$ SCLCs; tumors expressing high levels of MCL1, Bcl-X$_L$ and Bcl2 represent 8% of all SCLCs and mainly account for ASCL1$^+$ and NUROD1$^+$; finally, tumors expressing high levels of Bcl2 and low levels of MCL1 and Bcl-X$_L$ represent 4% of all SCLCs and account for POUF3$^+$ SCLCs [133].

Pelcitoclax (APG-1252) is a novel, dual Bcl2/Bcl-XL inhibitor being evaluated in combination with paclitaxel in patients with refractory/relapsed SCLC; preliminary findings

from the first-in-human study suggested a promising antitumor activity and a favorable safety profile [134].

MAPK pathway genes are frequently altered in NSCLC but are rarely altered in SCLC [135]. Genomic and proteomic studies indicate that MAPK activity is relatively suppressed in SCLC and is markedly lower than that observed in NSCLC. Interestingly, the SCLC-A subtype is selectively inhibited by MAPK activation in vitro and in vivo through induction of cell cycle arrest and cell senescence, while the NEUROD1 subtype was unaffected by MAPK activation. MAPK activation in SCLC-A cell lines resulted in strong upregulation of ERK negative feedback regulators and STAT signaling; the ERK-induced growth inhibition was independent of NOTCH [135]. These observations suggest a subtype-specific mitogenic vulnerability of SCLC [135].

### 6.5. Molecular Targets in MYC-Driven SCLCs

*MYC* is a major driver of SCLC development and evolution. A fundamental study by Mollaoglu et al. showed that Myc expression drives a non-neuroendocrine phenotype in *RB1⁻/TP53⁻* genetically modified mice and the developing tumors express NEUROD1 [18]. As above mentioned, *MYCL* is amplified or highly expressed in SCLC-A subtype and is required for its development [53]. In contrast, the other three SCLC subtypes, SCLC-N, SCLC-P and SCLC-Y, are associated with MYC amplification or overexpression. In a recent study, Ireland and coworkers have explored different SCLC mouse models with a time-series single-cell transcriptome analysis; they used two mouse models: MYC-driven SCLC with reduced expression of neuroendocrine markers and MYCL-driven SCLC with a high neuroendocrine entity [59]. Analysis in the time of neuroendocrine-positive tumors showed that a part of the tumor cells may undergo a transition to a non-endocrine phenotype, associated with MYC expression; in line with these observations, MYC exogenous expression can in neuroendocrine-positive SCLC induces a switch to a non-endocrine phenotype [59]. According to these findings it was proposed that MYC drives SCLC subtype evolution in vitro and that human SCLC subtypes correspond with MYC-driven evolution [58]. This de-differentiation process driven by MYC is triggered by NOTCH activation [58]. In line with these findings, human SCLC subtypes exhibit a consistent intramolecular subtype heterogeneity [59].

In line with these findings, another recent study by Patel et al., through the analysis of mRNA expression data of primary SCLC tumors and chromatin state profiling of SCLC cell lines, identified a L-Myc signature, enriched for neuronal pathways and a c-Myc signature enriched for a non-neuroendocrine transcriptional program, involving NOTCH signaling and epithelial to mesenchymal transition [136]. Genetic replacement of c-Myc with L-Myc in c-Myc-SCLC induced a neuroendocrine state but was unable to induce ASCL-SCLC. Functional analysis showed that c-Myc, but not L-Myc, was able to induce trans-differentiation promoting the transition from ASCL1-SCLC to NEUROD1-SCLC [136]. The trans-differentiation capacity of c-Myc is mediated through induction of REST, a neuronal differentiation repressor [136]. Importantly, the cells that replaced c-Myc with L-Myc became resistant to Aurora kinase inhibition [136]. These observations support the view that the plasticity between different histological and molecular subtypes of SCLC is regulated by c-Myc and L-Myc [136].

MYC is also an important determinant of drug sensitivity of SCLC. Mollaoglu et al. in a targeted drug screening showed that SCLC with high MYC expression is vulnerable to Aurora kinase inhibition [18]. An initial clinical study explored the therapeutic activity of Alisertib, an Aurora Kinase inhibitor, in various malignancies, including 60 patients with relapsed/refractory SCLC; objective responses were observed in 21% of these patients with a median duration of response of 4.1 months and a median PFS of 2.1 months [137]. Responses were observed both in platinum-sensitive and in platinum-resistant SCLCs [114]. A randomized phase II study of paclitaxel plus alisertib (an Aurora kinase inhibitor) versus paclitaxel plus placebo in relapsed/refractory SCLC patients showed no clinical benefit in the whole population of enrolled SCLC patients, but supported a potential benefit related to

c-Myc expression [138]. In fact, among 140 patients enrolled in the study with an evaluation of genetic alterations, patients with cell cycle regulator mutations and c-Myc expression showed an improved progression-free survival in patients treated with paclitaxel plus alisertib compared to those treated with paclitaxel plus placebo [138]. Particularly, in c-MYC-positive patients, representing 72% of cases, alisertib favored outcomes over placebo: median progression-free survival was 4.6 months with alisertib plus paclitaxel compared to 2.2 months with placebo and paclitaxel; conversely, in tumors c-MYC-negative, outcomes favored the placebo group, compared to the alisertib group (5.1 months vs. 3.3 months, respectively) [138]. The finding of better outcomes for patients with cell cycle regulator mutations in this study is not surprising in view of the observation that SCLC cells that lack *RB1* are dependent on aurora B kinase for their survival [139].

Interestingly, some patients with SCLC displaying MYCL1 fusion proteins reported deep and prolonged responses first to Aurora A Kinase inhibitor and subsequently to immune check inhibitors [140].

A methylome analysis carried out on a large panel of SCLC cell lines provided evidence about a potentially relevant association between TREX1, a 3′ exonuclease I that acts as a STING antagonist in the regulation of a cytosolic DNA-sensing pathway: increased TREX1 methylation and low expression of TREX1 were associated with the response different Aurora kinase inhibitors [75,141].

Another study provided evidence that the type of *MYC* family gene expressed in SCLC tumors is a major determinant of apoptotic sensitivity. Thus, Dammert et al., using CRISPR activation model, provided evidence that in contrast to *MYCN* and *MYCL*, *MYC* represses *BCL-2* gene transcription; the lack of BCL-2 expression determines a condition of sensitivity to cell cycle control inhibition and dependency on the expression of the anti-apoptotic protein Mcl-1 [142]. In line with this observation, an aurora kinase inhibitor with CHK1 inhibitor clearly prolongs the survival of mice bearing *MYC*-driven SCLC [142].

Chalishazar et al. have explored the metabolic profiles of different SCLC subtypes and found that MYC-driven SCLCs are preferentially dependent on arginine-regulated metabolic pathways, such as polyamine biosynthesis and mTOR pathway activation [143]. Both chemo-sensitive and chemo-resistant MYC-driven SCLCs are inhibited by arginine deprivation [143]. In mouse models of MYC-driven SCLCs, including patient-derived xenografts, were inhibited in their growth by arginine depletion induced by pegylated arginine deiminase (ADI-PEG 20). [143]. A recent study reported the results of a phase II study involving the administration of ADI-PEG 20 to 22 SCLC patients, subdivided into a chemotherapy-sensitive and a chemotherapy-resistant subgroups; the best overall response observed was stable disease in 2 patients in each cohort [144]. These patients were not stratified according to MYC status.

Other studies reported peculiar metabolic properties of MYC-driven SCLCs. Thus, Cargill et al. observed that MYC-overexpressing SCLCs displayed increased glycolysis gene expression directly driven by *MYC*, whereas SCLCs with low *MYC* expression were more reliant of oxidative metabolism; in line with these findings, inhibition of glycolysis with the PFK158 compound preferentially inhibited glucose uptake, ATP production, and lactate in MYC-overexpressing SCLC cell lines, resulting in an inhibition of their growth and survival [145]. MYC-overexpressing SCLCs, such as the POU2F3 subtype, displayed increased fatty acid metabolism [146], apparently triggered by the MEK5-ERK5 kinase axis [147].

Two recent studies clarified the biochemical mechanisms through which MYC stimulates both metabolism and protein synthesis. In fact, it was shown that MYC induces the expression of the guanosine biosynthetic enzymes inosine monophosphate dehydrogenase-1 and -2 (IMPDH1 and IMPDH2), two transcriptional targets of MYC, and, through this mechanism, activates guanosine triphosphate (GTP) synthesis [148]. This biochemical mechanism rendered MYC-expressing SCLCs dependent on IMPDH and this is a targetable vulnerability in chemoresistant MYC-high SCLC [149].

*6.6. Molecular Targets in SCLCs with MYCN Amplification*

*MYCN* gene amplifications were reported in about 4–6% of primary SCLCs [8,150,151]. However, *MYCN* amplification was not associated with a peculiar SCLC phenotype and the role itself of *MYCN* in SCLC remains unclear. Recent studies have better defined the possible role of *MYCN* in SCLC development and have also detected some potential therapeutic vulnerabilities. Grunblatt and coworkers have developed a mouse model of *MYCN*-driven SCLC, using an inducible MYCN-overexpression system [152]. The transcriptional changes induced by *MYCN* overexpression consisted in upregulation of MYC targets and of members of the unfolded protein response, in association with downregulation of immune signaling pathways [152]. *MYCN* overexpression in SCLC resulted in a marked chemoresistance to chemotherapy [152]. A CRISP-Cas9 RNA screen at genome-scale allowed to identify the deubiquitinase ESP7 as a tumor-specific vulnerability associated with *MYCN* overexpression [152]. Importantly, pharmacological inhibition of USP7 restored a chemosensitivity in chemoresistant *MYCN*-overexpressing PDX models, thus indicating a potential strategy for the treatment of these tumors [152].

Another study confirmed the chemoresistance of *MYCN*-expressing SCLC cell lines; MYCN promoted chemoresistance of these cells through an inhibition of drug-mediated apoptosis [153]. MYCN induced an increased expression of HES1 through a transcriptional mechanism and activates the NOTCH pathway [153]. In patient SCLC samples there was a positive direct correlation between MYCN mRNA levels and HES1 mRNA levels; at IHC level, MYCN expression was more frequently observed in chemorefractory (55%) than in chemoresponsive (10%) SCLCs; high MYCN levels were associated with poor survival [153].

Wang et al. reported a synergistic interaction between JQ1 (a BET inhibitor) and a BCL2 inhibitor (ABT-263) in inducing apoptosis of *MYCN*-amplified SCLC [154].

*6.7. Insulin Growth Factor Receptor 1 (IGF-1R) Targeting*

Recent studies have shown an important role of IGF-1R signaling in supporting the survival and proliferation of POU2F3 and ASCL1-driven SCLC.

A recent study characterized the role of the POU2F3 transcription factor in sustaining the development of a subgroup of SCLCs characterized by POU2F3 expression and low expression of neuroendocrine markers [55]. A CRISPR-screening of these tumors oriented to evaluate their dependency on kinase activities provided evidence that POU2F3 SCLCs are dependent on IGF-1R-mediated signaling through PI3K pathway [55]. In line with this finding, POU2F3 SCLCs were more inhibited by IGF-1R inhibitors than ASCL1 or NEUROD1-positive SCLCs [55].

Wang et al. have provided evidence that ASCL1[+] SCLCs could represent another SCLC subtype sensitive to IGF-1R inhibitors [131]. They have initially screened the secretome profile of various SCCLC subtypes and found that IGFBP5 is preferentially secreted by ASCL1[+] SCLCs [155]. IGFBP5 is a secreted protein that binds IGF-1 and prevents the interaction with its receptor IGF-1R [155]. This finding suggested that ASCL1 in addition to its pro-survival effect, induced also the production of IGFBP5, thus limiting IGF-1R signaling and its proliferative effects [155]. ASCL1[+] SCLCs are sensitive to BET inhibitors, such as JQ1; this compound inhibits ASCL1 expression by abrogating the interaction between BRD4 and the ASCL1 enhancer and in consequence of this inhibition of ASCL1 expression, it reduces IGFBP5 secretion, thus increasing IGF-1R-dependent signaling [155]. This finding supported the rationale for evaluating the combination of a BET inhibitor with an IGF-1R inhibitor: this combination of inhibitors resulted in a synergistic antitumor activity in ASCL1[+] SCLC cell lines [155].

Other studies have shown that IGF-1R was expressed in 53% of primary SCLCs and reported an inhibitory effect of a monoclonal antibody anti-IGF-1R in synergism with cisplatin and irinotecan [156].

A recent study showed that the TKI brigatinib, well known for its capacity to inhibit ALK, is also potent inhibitor of IGF-1R signaling [157]. Treatment of IGF-1R[+] SCLC cell

lines with brigatinib significantly increased their sensitivity to the cytotoxic effects of etoposide [157]. These observations support clinical trials in SCLC patients combining chemotherapy with brigatinib.

IGF-1R activates PI3K signaling and through this pathway inhibits apoptosis and promotes proliferation of SCLC. Insulin receptor substrate 1 and 2 (IRS1 and IRS2) are the main signal transmitters, involved in IGF-1R and insulin receptor signaling. A recent comprehensive genomic analysis on 73 SCLC samples identified 5% of cases bearing focal *IRS2* gene amplifications [158]. Generation of PDX from these tumors allowed exploring their pharmacological vulnerability, showing sensitivity to the TKI ceritinib, thus suggesting the opportunity of a targeted therapy for these tumors [159].

### 6.8. BET Inhibitors in SCLC Therapy

BET (bromodomain and extra-terminal) proteins interact with acetylated histones and recruit protein complex, thus regulating gene transcript BET proteins modulate the expression of MYC as well as of other genes with key role in oncogenic transformation. Lenhart et al. showed that SCLC cells are sensitive to growth inhibition by BET inhibitors, such as JQ1. JQ1 treatment of SCLC cells does not affect MYC levels and decreases ASCL1 expression [160]. JQ1 disrupts the interaction of BRD4 at the level of ASCL1 enhancer [160].

JQ1 decreases MYCL levels and induces apoptosis of SCLC cell lines with *MYCL* gene amplification or expression; however, there was no association between MYCL levels and sensitivity to JQ1 [161]. CDK6 expression and the extent of its reduction following treatment with JQ1 are associated with JQ1 sensitivity [161].

A consistent number of BET inhibitors are under evaluation in clinical trials in various types of hematological and solid tumors [162]. However, the clinical success using these agents in monotherapy was limited [163]. Few SCLC patients were included in some phase I clinical trials, showing no responses when these agents were used in monotherapy [163].

The antitumor activity of BET inhibitors was more promising when combined with other antitumor agents, as supported by various preclinical studies.

Lam et al. reported that about 50% of SCLC cell lines are sensitive to BET inhibitors, with induction of an apoptotic response [164]. Importantly, treatment of SCLC cells with BET inhibitors induced a decrease of the antiapoptotic proteins Bcl2 and BCLX$_L$; the BCL2 inhibitor Venetoclax synergizes with a BET inhibitor to induce apoptosis of SCLC cells [165]. A clinical study (NCT02391480) involving the combination of the BET inhibitor ABBV-075 with Venetoclax in patients with cancers, including SCLC, is ongoing.

Other studies have supported a synergism between BET inhibitors and PARP inhibitors. Bian et al. have explored PARP1 expression in SCLC, showing that high PARP1 expression was associated with a better overall survival [166]. In SCLC, *PARP1* expression was regulated at transcriptional level by *MYC*, *MYCL* and *MYCN*; in line with these findings, in SCLC tumor specimens there was an association between high MYC expression and high PARP1 expression [166]. Targeting of the BET-PARP1 axis using the combination of a BET inhibitor with a PARP1 inhibitor resulted in a synergistic antitumor effect [165]. Another recent study showed that BET inhibitors synergize with PARP inhibitors in inducing apoptosis of SCLCs [167]. Combinatorial efficiency was observed both in SCLC cells with *MYC*-amplified and *MYC*-WT SCLCs at an extent higher than in SCLC cells with impaired MYC signaling [167].

ASXL3 is a pluripotency transcription factor for respiratory epithelial cells [168]; this transcription factor determines a link between BRD4 and BAP1 forming a complex with BRD4 in the SCLC-A subtype, expressing ASCL1 [169]. Pharmacological inhibition with a BET-specific chemical degrader (dBET6) inhibits the growth of SCLC-A subtype cells [169].

A recent study reported the characterization of a new BRD4 degrader, CFT-2718 displaying a consistent efficacy in inducing degradation of BRD4 and tumor cell apoptosis [170].

*6.9. Netrin-3, a Potential Therapeutic Target for SCLC*

Cellular navigation cues are promoted by families of molecules, such as netrins, semaphorins and ephrins; these molecules play a key role in different cellular processes and their deregulation is frequently observed in cancer. A recent study provided evidence that Netrin-3 expression is deregulated in SCLC. Most SCLC primary samples express high levels of Netrin-3 and only rarely of Netrin-1 [171]. Netrin-3 expression in SCLC is promoted by ASCL1 and NEUROD1 [171]. In vitro and in vivo SCLC models supported the efficacy of the monoclonal antibody NP137, targeting both Netrin-1 and Netrin-3 and blocking their binding to their receptors, in inhibiting the growth of SCLC cells [171]. These observations support future clinical studies aiming to evaluate NP137 (an antibody that is under clinical evaluation) in monotherapy or in combination with chemotherapies or immunotherapies for the treatment of SCLC.

*6.10. Immunotherapy*

Immune checkpoint inhibitors (ICIs) targeting programmed cell death protein-1 (PD-1) or programmed death-ligand 1 (LD-L1) (or cytotoxic T-lymphocyte associated protein 4 (CTLA-4) have revolutionized the treatment of several cancer types, including NSCLC. Unfortunately, the clinical response is limited only to about 20% of NSCLC patients with advanced disease. Currently, the anti-PD-1 agent pembrolizumab is approved for use in first- and second-line therapy in patients with advanced NSCLC whose tumors are positive for PD-L1 expression by immunohistochemistry; nivolumab (anti-PD-1) and atezolizumab (anti-PD-L1) are both indicated for second-line therapy independently of PD-L1 expression. Recent studies have reported the clinical use of immunotherapy with checkpoint inhibitor monoclonal antibodies blocking PD-1, PD-L1 and CTLA-4, as single agents or in combination for the treatment of SCLC patients. These studies involved either the use of these agents in third line of treatment or in combination with chemotherapy in first line of treatment for newly diagnosed patients with extensive disease.

Concerning the use of ICIs as first-line treatment options, several studies have explored these drugs in combination with chemotherapy in SCLC patients. Two studies explored the efficacy of ipilimumab (an anti-CTLA-4 monoclonal antibody). A first phase II study randomized 130 patients to receive either carboplatin/paclitaxel plus ipilimumab or carboplatin/paclitaxel plus placebo; ipilimumab was administered in a phased or concurrent schedule [172]. The results of this first study supported the view that ipilimumab could be beneficial in a subset of patients [172]. However, a subsequent phase III study failed to show any significant difference in overall survival in 1132 SCLC patients randomized to receive platinum/etoposide plus ipilimumab or platinum/etoposide plus placebo [173].

The phase III randomized Impower133 study explored the efficacy of atezolizumab (an anti-PD-L1 monoclonal antibody) in treatment-naïve SCLC patients with extensive disease status; in this study, the patients received either atezolizumab plus platinum/etoposide or placebo plus platinum/etoposide: the overall survival was 12.3 months in the atezolizumab group and 10.3 months in the placebo group, while progression-free survival was 5.2 months and 4.3 months, respectively, at 18 months and the overall survival was 34% in the atezolizumab group and 21% in the placebo group [174,175].

The CASPIAN trial involved 805 treatment-naïve SCLC patients with extensive disease status, randomized to receive treatment with platinum/etoposide or platinum/etoposide plus durvalumab (an anti-PD-L1 monoclonal antibody) or platinum/etoposide plus durvalumab and tremelimumab (an anti-CTLA-4 monoclonal antibody) [176]. An interim analysis showed that the overall survival was longer in the durvalumab-chemotherapy arm compared to chemotherapy alone [176]. An update of this study presented at the 2020 American Society of Clinical Oncology (ASCO) [177] and then published on Lancet Oncology [178], with a median follow-up of more than 2 years showed an improvement of overall survival for durvalumab plus chemotherapy compared to chemotherapy alone (12.9 months versus 10.5 months); however, the addition of tremelimumab to durvalumab did not improve overall survival [177,178]. An updated OS analysis of this trial with

a median follow-up of >3 years showed that treatment with platinum/etoposide plus durvalumab demonstrated sustained OS benefit over platinum/etoposide treatment, with 3 times more patients estimated to be alive at 3 years when treated with EP+D versus EP alone (17.6% vs. 5.8%, respectively) [179].

The anti-PD-L1 monoclonal antibody was evaluated in combination with standard chemotherapy (platinum/etoposide) in 453 SCLC in the context of the randomized KEYNOTE-604 trial. The final analysis of the study showed an improvement of the progression-free survival in the arm treated with pembrolizumab chemotherapy, compared to chemotherapy alone: at 12 months, 45% vs. 39% and at 18 months 22.5% vs. 11.2%, respectively [180].

The study ECOG-ACRIN EAS161, a phase II randomized study carried out in 160 patients with SCLC with extensive disease evaluated the efficacy of platinum/etoposide plus nivolumab (an anti-PD-1 monoclonal antibody), compared chemotherapy alone: a higher progression-free survival (5.5 months vs. 4.7 months) and overall survival (11.3 months vs. 8.5 months) was observed in the nivolumab group compared to the chemotherapy alone group [181].

Concerning the use of ICIs as third-line treatment options, the two PD-1 inhibitors, nivolumab and pembrolizumab, were investigated in the context of three different clinical studies: the phase I/II CheckMate 032 clinical trial, exploring the efficacy of nivolumab alone or in combination with ipilimumab (anti-CTLA-4); the phase Ib study KEYNOTE-028 and the phase II study KEYNOTE-158 evaluated the efficacy of pembrolizumab for pretreated patients with PD-L1-positive tumors [182,183]. In the KEYNOTE-028 and KEYNOTE-158 studies the patients received two or more lines of previous therapy for SCLC; the two studies involved a total of 83 patients; the overall response rate was 19%, with two patients with complete response and 14 patients with partial response; 61% of responders had responses lasting 18 months or longer [182,183]. These studies showed some clinical efficacy, with a limited number of responding patients (ranging from 11% to 33%) and with a median improvement of progression-free survival ranging from 1.4 to 2 months [182,183].

The CheckMate 451 study evaluated Nivolumab and Ipilimumab versus placebo as maintenance therapy in SCLC patients with extensive disease; the study enrolled 834 patients and did not show any significant prolongation of overall survival compared to placebo [183]. A trend toward an overall survival benefit with treatment based on Nivolumab and Ipilimumab was observed in patients with a high tumor burden, defined as ≥13 mutations per megabase [184].

Since only a limited number of SCLC patients respond to ICIs, the identification of biomarkers predicting response is critical. Previous studies carried out in NSCLC patients have shown that the level of PD-L1 expression in primary tumor site or metastatic sites is predictive of the response to the treatment with ICIs [185]. The identification of biomarkers predictive of the response of SCLC patients to ICIs treatment remains difficult. A careful assessment of PD-L1 expression at protein and mRNA level in SCLC specimens was reported by Yu et al. by immunohistochemistry using two different anti-PD-L1 monoclonal antibodies and by in situ hybridization; predominant tumor-cell localized expression of PD-L1 was observed in 16.5% of cases and higher PD-L1 expression correlated with more infiltration of tumor-infiltrating lymphocytes [186].

The analysis of the tumor specimens in IMPOWER133, CASPIAN and KEYNOTE-604 clinical trials showed that PD-L1 expression ≥1% was observed in about 40% of SCLC patients and the levels of PD-L1 expression were not predictive of overall response rate, progression-free survival and overall survival [174,175,180].

SCLC is characterized by high somatic burden, in large part due to its association with smoking. Previous studies have shown the existence of an association in various tumors, including melanoma, NSCLC and urothelial carcinoma, between high tumor mutational burden and response to ICIs. Thus, Hellmann and coworkers explored the impact of tumor mutational burden on the efficacy of nivolumab alone or in combination with ipilimumab in patients with SCLC, in the context of the nonrandomized and randomized cohorts of

the CheckMate 032 study [187]. The results showed that the increased benefit to ICIs therapy, particularly for the combined therapy, correlates with high tumor mutational burden [187]. Particularly, in patients treated with nivolumab plus ipilimumab the median overall survival the overall survival was 3.6 months, 3.4 months, and 22 months in the low ($\leq$143 mutations), intermediate (143–247 mutations) and high ($\geq$248 mutations) mutations groups, respectively [187]. These results were in part confirmed by Ricciuti and coworkers who explored tumor burden by next generation sequencing in 52 SCLC patients treated with ICIs; they observed that no significant difference in the objective response rate between patients with a low or high tumor mutational burden was observed; the median progression-free survival and the median overall survival were significantly longer among patients with high tumor mutational burden compared to those with low tumor mutational burden [188]. However, the IMPOWER133, CASPIAN and KEYNOTE-604 trials, all involving the administration of ICIs in combination with chemotherapy, showed that tumor mutational burden was not predictive of a response to ICIs [176,177,182]. Thus, at the best TMB could be considered a potentially predictive factor for relapsed SCLC patients undergoing treatment with ICIs alone; furthermore, TMB does not seem to have a predictive value for SCLC patients undergoing treatment with ICIs and chemotherapy as first-line treatment. Prospective studies are required to carefully assess the role of mTMB to predict response to ICI.

Recent studies on gene expression analysis of SCLC subtypes suggest that a specific SCLC subtype could derive the greatest benefit from ICIs. Thus, Gay et al. reported an analysis of the patters of transcription factor programs and of immune pathway activation in SCLC, defining four major tumor subtypes [56]. In line with previous studies, differential expression of ASCL1, NEUROD1 and POUF3 transcription factors defined three SCLC subtypes, each characterized by the high expression of one of these three transcription factors; the fourth subtype was characterized by low expression of these three transcription factors and defined as SCLC-inflamed (SCLC-I) for the high expression of immune-related genes [56]. The gene expression pattern characterizes SCLC-I as a unique SCLC subtype: highest total immune infiltrate, with highest numbers among SCLC subtypes of CD8-positive T-lymphocytes, NK-lymphocytes and macrophages; higher expression than in other SCLC subtypes of genes encoding HLA and other antigen presenting machinery, as well as of interferon-$\gamma$-related T cell gene expression profile; higher expression of CD274 (PD-L1) and of PDCD1 (PD-1), as well as of CD80 and CD86, encoding the ligands for CTLA-4 [56]. Another typical feature of SCLC-I subtype is represented by the most mesenchymal features, characterized by low expression of epithelial markers such as E-cadherin and high level of mesenchymal markers, such as vimentin and AXL [56]. The analysis of the clinical response observed in the context of IMPOWER133 study at the level of different transcriptomic subsets of SCLC patients showed that SCLC-I tumors derive greater benefit from ICIs than other SCLC subtypes [56]. Particularly, this analysis showed that addition of pembrolizumab improved OS in all four SCLC subtypes, with a trend for a better response in SCLC-I compared to other subtypes; in SCLC-I tumors the median OS was 18 months among patients treated with atezolizumab compared with a survival of 10 months in patients treated with placebo [56]. Finally, through single cell analyses it provided evidence that the various SCLC subtypes display a consistent intratumor heterogeneity and that the development of cisplatin resistance was associated with the emergence of a cluster of cells displaying the properties of SCLC-I subtype, seemingly derived from SCLC-A cells that have undergone subtype switching, related to fluctuations in the level of NOTCH pathway activation [56].

Studies performed by Owonikoko have shown that the SCLC subtype characterized by YAP1 expression is associated with a T-cell inflamed phenotype [189]. These authors have analyzed 59 SCLC cases for the gene expression profile and observed that 16.9% can be classified as SCLC-A, 5.1% as SCLC-N, 10.2% as SCLC-Y, 6.8% as SCLC-P. Interferon-$\gamma$ pathways were found to be markedly upregulated in long-term survivors; a T-cell-inflamed profile was found to be enriched in long-term survivors [189]. According to

these findings, these authors have explored whether a validated 18-gene T cell-inflamed GEP (gene expression profile) signature, originally discovered in melanoma patients as a predictor of clinical response to PD-1 blockade [190] and validated through the analysis of patients treated with pembrolizumab in the context of the umbrella clinical trial KEYNOTE-028 [191], is differentially expressed in short-term and long-term SCLC survivors. This T-cell-inflamed signature was preferentially expressed in long-term SCLC survivors and its analysis in the various SCLC subtypes showed the highest expression in the SCLC-Y subtype [189]. SCLC-Y subtype, as well as long-term SCLC survivors are characterized by high expression of HLA gene family [167]. Finally, SCLC patients characterized as pertaining to the SCLC-Y subtype for their gene expression profile are associated with better prognosis [189].

The study of patients with limited-stage SCLC tumors (LS-SCLCs) offers the unique opportunity to explore tumoral tissue since these patients are suitable candidates for surgical resection. Thus, Chen and coworkers have explored intratumor heterogeneity and immunological competence of LS-SCLCs using whole exome/T cell receptor (TCR) sequencing and immunohistochemistry [192]. In this analysis, they compared LS-SCLCs with LS-NSCLCs, showing that LS-SCLCs have a degree of intratumor genetic heterogeneity and of predicted neoantigen burden comparable to that of NSCLCs, but significantly colder and more heterogenous TCR repertoire associated with higher chromosomal copy number aberration (CNA) burden [192]. Particularly, the analysis showed the existence of a particularly cold TCR repertoire at quantitative (density) and qualitative (richness and clonality) levels compared to normal lung tissue as well as to NSCLCs; in addition to this cold intratumor TCR repertoire, SCLCs showed also and markedly heterogeneous TCR repertoire with only 0.2–14.6% of all T cells identified across all tumor regions within the same tumors [192]. The cold and heterogeneous TCR repertoire may explain the limited response to immunotherapy of these tumors [192]. A remarkable difference between SCLCs and NSCLCs is that the former ones have a significantly higher CNA burden compared to the latter ones; in SCLCs, the CNA burden is negatively correlated with T-cell quantity (density) and quality (richness and clonality); in many other tumors high levels of CNAs correlate with reduced expression of markers of cytotoxic infiltrating immune cells and increased expression of tumor immune escape mechanisms [193,194].

Chan et al. have combined two techniques, single-cell RNA sequencing and multiplexed ion beam imaging, to explore the heterogeneity of tumors and of their immunological microenvironment [194]. The single-cell transcriptomic analysis showed that the consistent transcriptomic heterogeneity of SCLC contrasts with the uniform poor prognosis of patients [195]. An unsupervised clustering of SCLC malignant cell compartment identified 25 clusters. One of these clusters, cluster 22, was characterized by high expression of phospholipase C gamma 2 (*PLCG2*) gene. This gene drives a stem-like, pro-metastatic phenotype of this cluster recurrent in all SCLC subtypes [195]. Immunohistochemical analysis showed the presence of a small stem-like, pro-metastatic subpopulation with PLCG2 expression; the expression level of this tumor subpopulation had a negative prognostic impact. This study has also given the opportunity to evaluate the influence of SCLC subtype on the immune tumor microenvironment; this analysis was limited to the two most recurrent SCLC subtypes, SLC-A and SCLC-N [195]. This analysis showed that collectively decreased immune infiltrate was observed in SCLC, and particularly in the SCLC-N subtype; immune sequestration was observed in SCLC cases that do contain more immune cells [195]. Single-cell analysis of the cells of the tumor microenvironment showed that SCLC-A and SCLC-N subtypes display significant difference for their content of different T lymphocyte subpopulations, showing relative depletion of cytotoxic T cells and increase in Tregs in SCLC-N [195]. These studies showed also that a subpopulation of monocytes/macrophages displaying a profibrotic/immunosuppressive phenotype, was increased in its expression. In SCLC; these profibrotic monocytes/macrophages are particularly enriched in SCLC-N [195].

Roper et al. reported the results of an extensive genomic, transcriptomic, and proteomic analysis on 20 SCLC patients treated with durvalumab (anti-PD-L1) and olaparib;

4 of these 20 patients displayed a clear benefit related to treatment [195]. Genomic alterations did not correlate with clinical benefit to treatment; gene expression analysis showed strong enrichment of immune-related pathways and of NOTCH-signaling genes, both associated with a low neuroendocrine profile [196]. These findings were confirmed in an additional cohort of 36 SCLC patients treated with ICIs [196]. Multivariate analysis showed that NOTCH signaling was the only significant predictor of clinical benefit to ICIs; among NOTCH signaling target genes, the two gene showing markedly higher levels in patients responding to ICIs were *REST* and *NOTCH3* [196].

A recent study by Mahadevan et al. showed that tumor major histocompatibility complex I (MHC I) is a major determinant of clinical responses to ICIs; in fact, SCLC patients with high MHC I expression (15% of total) displayed a much higher survival than those with low MHC I expression following treatment with ICIs [197]. Most SCLCs (71%) displayed absent/low MHC I expression, while only 15% of these tumors displayed high MHC I expression; MHC I-high tumors displayed a low neuroendocrine phenotype, upregulation of epithelial-to-mesenchymal transition markers, upregulation of genes involved in MHC I antigen presentation (such as *TAP1*) and in IFN-γ signaling and increased infiltration of CD3+ and CD8+ lymphocytes and of lymphocytes expressing PD-L1 [197]. Antigen presentation defect observed in SCLCs expressing low levels of MHC I was related to epigenetic silencing of TAP1, a molecule involved in the machinery of antigenic presentation [197]. The study of experimental models supported a higher immunogenicity of non-neuroendocrine MHC I-high SCLCs; the immunogenicity of MHC I-low tumors can be stimulated by EZH2 inhibition [197].

Another approach to identify biomarkers associated with responsiveness of SCLC to treatments, including immunotherapy, consisted in analyzing the properties of long-term SCLC survivors (survival >4 years) compared to normal SCLC survivors (<2 years). Interestingly, this analysis showed that increased numbers of tumor-infiltrating lymphocytes were detectable in tumors of SCLC long-term survivors; the histological analysis of the tumors of long-term survivors showed also the existence of tumor areas enriched in immunosuppressive cells, such Treg, monocytes and macrophages, but the ratio of these immunosuppressive cells with respect to CD3+ lymphocytes were lower in long-term survivors of SCLC, thus suggesting the existence of a condition less immunosuppressive in these tumors [198]. Other studies on resected SCLC specimens have further supported that survival was more elevated among SCLC patients displaying in their tumors a higher number of T cells and B cells, whereas a high expression of PD-L1 at the level of tumor cells (observed in 3.2 of cases) or of tumor-infiltrating lymphocytes (observed in 33.5% of cases) correlated with worse survival [199].

Another recent study explored the immunogenicity of expression profiles in SCLC subtype and showed that: the NEUROD1 subtype displayed the lowest expression of immune-related genes, whereas the POUF23 subset exhibited the highest immunogenic profile; the ASCL1 subtype showed a wide spectrum of immunogenicity; NK and T cell scores were highest in the POUF23 subtype, whereas the ASCL1 subset was highly heterogeneous [200]. Recent studies suggest that the stimulation of natural killer (NK) activity could represent another strategy to potentiate the immune response against SCLC. The study of genetically engineered mouse models showed that the absence of NK cells, significantly enhanced the metastatic capacity of SCLC cells in vivo; stimulation of NK activity through administration of IL-15 reduced the metastatic activity of SCLC [200]. Another study showed that SCLCs evade innate immunity mechanisms mediated by NK lymphocytes through loss of loss of NK cells recognition of these tumors by reduction of NK-activating ligands (NKG2DL); in fact, primary SCLC tumors express low levels of NKG2DL mRNA [201]. Treatment of SCLC tumor cells with histone deacetylase inhibitors induced NKG2DL expression and promoted tumor infiltration by T and NK lymphocytes [201]. Thus, epigenetic silencing of NKG2DL in SCLC results in an immunosuppressive effect, limiting the tumor immune surveillance by NK lymphocytes.

Another area of active investigation consists in the identification, beyond PD-L1, of new inhibitory checkpoint ligands amenable to therapeutic targeting. Thomas et al. recently reported the screening of a large cohort of SCLC cell lines and primary tumor samples for the expression of validated inhibitory checkpoint ligands [202]. Among 39 identified checkpoint ligands, particularly remarkable seems the checkpoint protein B7-H6 exhibiting increased protein expression in SCLC relative to PD-L1; B7-H6 expression is associated with longer PFS and increased immune infiltrates in SCLC [202].

The poliovirus receptor (PVR, also called CD155) is an immune checkpoint molecule expressed on the surface of various tumor cell types. This molecule is able either to induce T cell activation through CD226 or to inhibit T cell activation via interaction with TIGIT. TIGIT is a competitor of CD226 for binding to CD155. CD155 is expressed at low levels in different types of normal epithelial cells but is overexpressed in several carcinomas. Recent studies have shown that CD155 is overexpressed in SCLC cell lines, with the highest expression at the level of cell lines isolated from pre-treated SCLC patients [203]. CD155 was significantly expressed in most SCLC primary tumors, its expression being observed on the membrane of tumor cells but not of immune cells present in the tumor microenvironment; high CD155 expression was associated with poorer prognosis but the difference with respect to negative tumors was not statistically significant [203]. Xu et al. have explored the expression of PD-L1, PD-1, CD155 and TIGIT in 60 patients with SCLC undergoing tumor surgical resection; high expression levels of CD155, PD-L1 or PD-L1 + CD155 were associated with a reduced overall survival [204]. High PD-1 or TIGIT expression are not associated with shorter overall survival [204]. Dora et al. have explored the expression of immune checkpoint molecules in SCLC subtypes and observed that (i) TIGIT was expressed in 75% of LS-SCLCs; and (ii) the expression of CD8[+] T-lymphocytes and of immunosuppressive molecules, such as CD155 and TIM.-3 was greater in NE-low than in NE-high SCLCs [205]. Lee and coworkers have explored the potential role of the polymorphisms of immune checkpoint related genes in the chemotherapy responses and survival outcome of SCLC patients [206]. The CD155 rs1058402G>A (Ala67Thr) polymorphism was associated with a worse chemotherapy response and overall survival [206].

The introduction of immune checkpoint inhibitors has revolutionized the therapy of SCLC demonstrating survival benefit when administered in combination with chemotherapy in patients with extensive stage disease, supporting this as the new standard of care. However, from these studies it emerges also that the extent of clinical benefit in SCLC is clearly lower than that observed in NSCLC [207]. This difference seems to be related to the presence of more immunosuppressive mechanisms operating in SCLC than in NSCLC and indicates the absolute need in future studies to better understand the immune anti-tumor mechanisms operating in SCLC [207]. Future studies will also clarify the potential role of ICIs in the treatment of SCLC patients with limited-stage disease as an adjuvant treatment after surgical resection and chemo-radiation. Ongoing clinical trials are evaluating ICIs in combination with other drugs, including PARP inhibitors, DLL3 targeting agents, fucosyl-GM1 monoclonal antibodies.

*6.11. Ferroptosis, a New Potential Therapeutic Target in SCLC*

Bebber et al. have explored cell death pathways in SCLCs [208]. They showed that treatment-naïve SCLCs present with inactivation of regulated cell death pathways, such as apoptosis and necroptosis [208]. The analysis of ferroptosis machinery in primary SCLC samples showed several interesting features [208]. Ferroptosis is an iron-dependent form of regulated necrosis, whose machinery is independent of the molecular components involved in apoptosis or necrosis; at morphological level, ferroptosis is characterized by consistent alterations in the mitochondrial function and structure, but not accompanied by rupture of cell membranes and nuclear shrinkage [208]. Interestingly, in SCLC cells it was observed a strong upmodulation, compared to normal lung tissue, of the levels of the cysteine/glutamine transporter SLC7A11 [208]. SLC7A11 is an antiporter importing cysteine and exporting glutamate. The inhibition of uptake of cysteine into cells deter-

mines reduced levels of glutathione and triggers ferroptosis. Interestingly, the analysis of the response of a panel of SCLC cell lines to a ferroptosis challenge showed that these cell lines can be subdivided in responders and non-responders; particularly, SCLC with non-neuroendocrine phenotype were sensitive to ferroptosis, while SCLC cell lines with a neuroendocrine phenotype are resistant [208]. Non-NE and NE SCLC subtypes displayed a different lipid remodeling, responsible for the differential sensitivity to ferroptosis triggering (non-NE SCLC subtypes are sensitive to ferroptosis for the upregulated ether lipid biosynthesis) [208]. NE-SCLC subtypes are resistant to ferroptosis for a reduced capacity of synthesis of endogenous reduced glutathione (GSH); however, these calls may be rendered sensitive to ferroptosis by thioredoxin (TRX) inhibitors that increase GSH levels [208]. Thus, combined ferroptosis induction and TRX pathway inhibition resulted in a marked anti-tumor activity in vitro and in vivo in NE-SCLC subtypes [208]. These observations support the development of therapeutic approaches based on the differential sensitivity of SCLC subtypes to ferroptosis.

Combined SCLC (C-SCLC) is composed of SCLC admixed with non SCLC tumor components, represented either by adenocarcinoma (Co-ADC), LCNEC (Co-LCNEC) or squamous cell carcinoma (Co-SQC); the analysis of ferroptosis sensitivity/resistance markers showed that Co-SQC, exhibiting a non-NE transcriptional profile, was classified as sensitive, Co-LCNEC, exhibiting a NE transcriptional profile, was classified as resistant and Co-ADC, characterized by mixed NE and non-NE transcriptional profile, by a heterogeneous pattern of sensitive/resistance [209].

Sulforaphane, an isothiocyanate compound derived from cruciferous vegetables (broccoli sprouts), induces in vitro cell death of SCLC cells via induction of ferroptosis through targeting of the cysteine/glutamate antiporter SLC7A11 [210].

### 6.12. CDK4 and CDK6 Inhibitors

To improve the clinical efficacy of chemotherapy and of ICIs another possible approach consisted in using drugs that may both lower the toxicity of chemotherapy and improve the antitumor immune response induced by ICIs. One candidate drug with this pharmacological profile is trilaciclib, a CDK4/CDK6 inhibitor that maintains G1 cell cycle arrest of cells that are dependent on CDK4/CDK6 for regulation of G1 to S transition; this drug by transiently maintaining hematopoietic stem cells in G1 arrest during chemotherapy protects them from chemotherapy-induced damage and thus reduces hematopoietic toxicity induced by chemotherapy [211]. In experimental mouse models, the addition of trilaciclib to chemotherapy and ICI combinations enhanced antitumor response compared with chemotherapy alone, by modulating T-cell proliferation and composition in the tumor microenvironment; the efficacy of anti-PD-L1 antibody was not increased by trilaciclib administration, in the absence of chemotherapy [211]. The potentiation of immune response elicited by trilaciclib seems to be related to an increased effector function in the tumor microenvironment, as supported by the presence of elevated proportions of CD4$^+$/CD8$^+$ lymphocytes [211].

In SCLC patients, trilaciclib reduced the myelotoxicity and myelosuppressive effects of chemotherapy but did not improve PFS and OS [212]. Three phase II, double-bind, placebo-controlled, clinical studies have shown that the administration of trilaciclib prior to chemotherapy regimens used in the first- or second-/third line of treatment significantly improved chemotherapy-induced myelosuppression and quality of life of SCLC patients, with no impact on the antitumor efficacy [213].

Another clinical trial evaluated the effects of administering trilaciclib prior to chemotherapy and atezolizumab to newly diagnosed SCLC patients with extensive-stage disease. Compared with the placebo, trilaciclib improved chemotherapy-induced myelosuppression but did not modify antitumor efficacy outcomes [214]. Interestingly, administration of trilaciclib compared to placebo induced more newly expanded peripheral T lymphocyte clones, with a more pronounced expansion observed in patients exhibiting an antitumor response to the treatment with atezolizumab + chemotherapy [215].

Trilaciclib was approved by FDA in February 2021 for preservation of bone marrow function in patients with extensive SCLC undergoing chemotherapy treatment.

*6.13. Fucosyl-GM1*

In 1986, Nilsson et al. reported the expression of a ganglioside antigen, Fucosyl GM1 on the surface of SCLC [216]. The expression of Fucosyl GM1 in SCLC is due to the presence in these tumor cells of FUT1 and FUT2, two GM1 synthases [217]. Specific monoclonal antibodies were used to detect the expression of Fucosyl GM1 in SCLC, showing positivity in 68% of primary tumors [217]. One of these antibodies, BMS-986012, was selected for clinical development for its properties: this fully human IgG1 antibody specifically binds to Fucosyl GM1 with high affinity; the binding of BMS-986012 to Fucosyl GM1-expressing SCLC resulted in enhanced antibody-dependent cellular cytotoxicity; in several mouse models, BMS-986012 showed a consistent efficacy and was well tolerated; the combined administration of BMS-986012 with an anti-PD-1 monoclonal antibody resulted in enhanced anti-tumor activity [217]. Interestingly, Lee et al. have explored the potential role of the polymorphisms in immune checkpoint-related genes in the chemotherapy responses and survival outcomes of SCLC patients [206]. The CD155 rs1058402 G>A (Ala67Thr, A67T) was associated with a worse chemotherapy response and overall survival [206]. This CD155 mutation (Ala67Thr) increases the binding affinity for and the signaling via an inhibitory immunoreceptor TIGIT [218] These observations have supported phase I/II clinical studies using BMS-986012.

The initial results of a phase I/II study (NCT 02247349) enrolled SCLC patients with refractory/relapsed SCLC undergoing treatment with BMS-986012 (at doses from 70 to 1000 mg every 3 weeks) [219]. In 29 treated patients 1 CR and 1 PR were observed; 4 patients displayed stable disease [219]. In an ongoing phase I/II clinical trial, the combination of BMS-986012 and Nivolumab elicited 38% of objective responses among relapsed/refractory SCLC patients, a finding favorable with respect to a historical control of Nivolumab monotherapy (12%) [220]. BMS-986012 is under evaluation in combination with chemotherapy and Nivolumab in the frontline treatment of advanced SCLC (NCT 047702880). Another ongoing clinical trial (NCT 02815592) is evaluating BMS 02815592 in combination with platinum/etoposide as first-line treatment of SCLC patients with extensive disease [221]. A recent report showed that this therapeutic regimen is associated with a safety profile like that historically reported in patients undergoing treatment with platinum and etoposide alone [221].

## 7. Patient Derived Xenografts: A Fundamental Tool for Therapeutic Development in SCLC

Patient-derived xenograft (PDX) models represent a fundamental tool to develop SCLC preclinical models and to predict drug sensitivity. PDX models are generated by purifying tumor cells from a patient tumor and engrafting these cells into immunodeficient mice, without in vitro passages [222]. PDX models have given a consistent contribution to define the drug sensitivity of SCLCs to BCL2 inhibitors, arsenic trioxide, PARP inhibitors, topoisomerase inhibitor 1, to study metabolic alterations in SCLCs; furthermore, the study of PDX has also given a fundamental contribution to correlate clinical response with responses in corresponding PDX models at the level of individual patients [222]. The two most consistent limitations of PDX models are that (i) not all primary patient tumors generate xenografts; and (ii) xenograft development is conditioned by tumor aggressiveness and mouse strain used to grow in vivo tumors [222].

The fundamental role of PDX in the study of SCLC intertumoral and intratumoral heterogeneity was strongly supported by several recent studies. Thus, Drapkin and coworkers have explored the generation of PDX from SCLC tumor biopsies and CTCs: PDXs were obtained with at 89% efficiency from tumor biopsies and 38 from CTCs [223]. Whole-exome sequencing studies showed that somatic mutations are stably maintained between patient tumors and PDXS; importantly, early-passage PDXs maintained the genomic and transcrip-

tional profiles of founder PDXs [223]. Similar conclusions were reached by Simpson and coworkers who reported the development of a biobank of 38 PDXs, isolated from CTCs with an efficiency of 40%; in some instances, PDXs were isolated at diagnosis and at relapse and are a suitable model to explore the mechanisms of drug resistance [57]. Vickers et al. have explored 147 SCLC patients for CTC-derived explants (CDXs) and observed CDXs were generated from 34 patients; several peculiar features were associated with positive generation of CDXs: (i) CTC number was significantly higher in blood samples which successfully generated a CDX from those which did not; (ii) patients generating CDXs display a higher proportion of chemorefractory cases; and (iii) patients generating CDXs have a shorter progression-free survival and overall survival [224].

The single cell-analysis at the level of CTCs and of CTC-derived xenografts (CDX) allowed to define the level of genetic heterogeneity of SCLC and its contribution to therapeutic resistance [56]. CTCs were derived from patients at different time points during therapy. In both cellular models the onset of drug resistance, was associated with an increase of intratumor resistance; longitudinal single-cell profiling of CTCs derived from patients before and after platinum relapse showed increased intra-tumor heterogeneity after relapse in association with unique gene expression patterns that were reproduced in paired CDXs [56]. These observations support the need to potentiate the efficacy of frontline treatments of SCLC before these tumors can develop consistent intra-tumor heterogeneity.

## 8. Other Neuroendocrine Lung Tumors

Neuroendocrine tumors (NET) of the lung comprise a heterogeneous group of tumors that differ for their molecular and biological properties and exhibit different degrees of malignancy, ranging from indolent lesions associated with long-term life expectancy (low-grade cell differentiated bronchial carcinoids, known as atypical carcinoid) to aggressive tumors associated with poor outcomes (such as large cells neuroendocrine carcinomas and SCLCs) [225]. The 2015 World Health Organization (WHO) classification has grouped four histological variants of NETs: typical carcinoid (TC) and atypical carcinoid (AC) as low-grade carcinoids; large cell neuroendocrine carcinoid (LCNEC) and SCLC as high-grade carcinoids [226,227]. The main properties of these tumors are summarized in Table 5. At the clinical level, TCs are low-grade tumors associated with a good prognosis and successfully treated with surgery alone; ACs are intermediate-grade tumors associated with a more aggressive clinical course and are treated with a multimodal approach based on chemo-radiotherapy; LCNECs and SCLCs are high-grade tumors associated with a very aggressive clinical course and with a dismal prognosis [228]. The various NETs are distinguished to cytological and immunohistochemical criteria: TCs and ACs display a well-differentiated neuroendocrine morphology being composed by cells resembling normal neuroendocrine cells present in pulmonary epithelium, exhibit a organoid growth pattern, not have necrotic areas, express neuroendocrine markers and have a mitotic index <10 mitoses/2 mm$^2$; LCNECs and SCLCs are composed by poorly differentiated neuroendocrine cells, display variable growth pattern (trabecular, solid and diffuse), exhibit numerous necrotic areas, express neuroendocrine markers and possess a mitotic index >10 mitoses/2 mm$^2$ [228,229]. The criteria and terminology of lung cancer neuroendocrine tumors in the 2021 WHO classification of thoracic tumors (5th edition; 2021) remained basically unchanged from the previous edition [229]. The most important parameter for distinguishing typical/atypical carcinoids from SCLC and LCNEC remained the number of mitotic counts per 2 mm$^2$ [230].

**Table 5.** Main biologic properties and genetic abnormalities of different types of lung neuroendocrine tumors.

| Tumor | % of Lung Tumors | Main Biologic Properties | Recurrent Genetic Alterations | Molecular Subgroups | Prognosis Survival |
|---|---|---|---|---|---|
| Typical Carcinoid | 1–2 | Cell size variable Low mitotic index Ki-67: low | Mutation/loss: EIF1AX, ARID1A, LRP1B, NF1, DST | | Good 5-yr: high |
| Atypical Carcinoid | 1–2 | Cell size variable Low/intermediate mitotic index Ki-67: intermediate | Mutation/loss: MEN1, ATP1A2, EIF1AX, ARID1A, SMARCA4, PKD1, AMER1, RAD51C | Cluster A1: immune infiltration, ASCL1$^+$, DLL3$^+$ Cluster A2: EIF1AXmutation SLIT1, ROBO1 downregulation Cluster B: MEN1 mutations, monocytes, poor prognosis. | Variable 5-yr: middle |
| Large-cell NE cancer (LCNEC) | 3 | Cell size large, with abundant cytoplasm High mitotic index Ki-67: high | Mutation/loss: TP53, LRP1B, RB1, SYNE1, ADAMTS12, USH2A, KEAP1, STK11, PTEN, NOTCH1, KMT2A Amplification: MYC, MYCN, MYCL1 | Type I (TP53, STK11, KEAP1) Type II (TP53, RB1) | Poor 5-yr: low |
| Small-cell lung cancer (SCLC) | 15–20 | Cell size small, with scarce cytoplasm High mitotic index Ki-67: very high | Mutation/loss: TP53, RB1, LRP1B, CSMD3, ZFHX4, SYNE1, USH2A, KMT2D, PTEN, NOTCH1, KMT2A Amplification: MYC, MYCN, MYCL1 | SCLC-A SCLC-N SCLC-P SCLC-Y | Poor 5-yr: very low |

Consistent differences at the level of biologic and molecular features allow to distinguish low-grade neuroendocrine carcinomas (LGNECs) from high-grade neuroendocrine carcinomas (HGNECs): tumor mutation burden is markedly higher in HGNECs than in LGNECs; *TP53* and *RB1* mutations are frequent in HGNECs, but are rare in LGNECs; smokers are very frequent among HGNECs, but are rare in LGNECs; LGNECs display frequent alterations in chromatin-remodeling genes such as *MEN1*, *ARID1A* and components of the SWI/SNF complex; paraneoplastic syndromes are frequent in HGNECs but rare in LGNECs; 13q and 17p chromosome deletions are frequent in HGNECs, but absent in LGNECs, while 11q chromosome deletions are common to all neuroendocrine lung tumors; amplifications of *MYC* family genes are frequent in HGNEC but absent/rare in LGNEC; decrease of E-cadherin expression, alterations of the Rb pathways and defects of the extrinsic and intrinsic apoptotic pathway are found in HGNECs, but absent in LGNECs; *FHIT* tumor suppressor is inactivated in HGNECs, but not in LGNECs [229].

Various studies have contributed to the identification of the most recurrent genetic alterations observed in the different types of NETs. In 2014, Fernandez-Cuesta and coworkers, performed an integrated genome/exome sequencing of 44 LGNECs. Covalent histone modifiers and subunits of the SWI/SNF complex are mutated in 40% and 22% of cases, respectively [231]. *MEN1*, *ARID1A* and *EIF1AX* are the most recurrently mutated genes; *ARID1A* and *MEN1* genes play an important role in chromatin remodeling [231]. Mutations of the eukaryotic translation initiation factor 1A (*EIF1AX*) were observed in 9% of cases [186]. Other gene mutations observed in these tumors occur at the level of histone methyltransferases (*SET1B*, *SETDB1* and *NSD1*), demethylases (*KDM4A*, *PHF8* and *JMJD1C*) a in members of the Polycomb complex (*CBX6*, *EZH1* and *YY1*). *TP53* and *RB1* gene alterations were only very rarely observed in these patients. The presence of only

these mutations affecting few biochemical pathways and the virtual absence of other cancer-associated mutations support an independent origin of these tumors from HGNECs [231].

*MEN1* is a tumor suppressor gene that encodes the protein menin, mutated at high frequencies in neuroendocrine tumors. *MEN1* acts as an epigenetic regulator and its genetic deficiency is associated with the inherited tumor syndrome multiple endocrine neoplasia type 1 (MEN1). The study of a mouse model driven by mutant *KRAS* and homozygous deficient *MEN1* leads to neuroendocrine differentiation of lung cancer: deficiency of menin resulted in the accumulation of DNA damage and antagonized oncogenic *KRas*-induced senescence [232].

More recently, molecular studies comparatively analyzed the molecular abnormalities of the four different subtypes of neuroendocrine tumors and better defined the molecular heterogeneity of each of these subtypes. Thus, Simbolo and coworkers have performed a whole-exome sequencing analysis 148 NETs, corresponding to the four subtypes of lung neuroendocrine tumors (including 53 TCs, 35 ACs, 27 LCNECs and 33 SCLCs), showing their peculiar clinical-biological and molecular properties [232]. At clinical/biological level, the mean age of patients with carcinomas was higher than that of patients with carcinoids; the proportion of patients with high tumor stages (III and IV) progressively increased from TCs, to ACs, LCNECs and SCLCs; the proportion of smokers was higher in carcinomas than in carcinoids; mitotic count and Ki67 index progressively increased from TCs to SCLCs [233]. Carcinomas had more mutations than carcinoids: TCs 0.8/Mb, ACs 1.6/Mb, LCNECs 3.0/Mb and SCLCs 2.9/Mb [233]. Some genes, such as those involved in the control of cell cycle (*TP53*, *RB1* and *ATM*) and *KMT2D* are preferentially mutated in carcinomas, while other genes such as *MEN1* are preferentially mutated in carcinoids [233]. In contrast, genes involved in the control of chromatin remodeling, such as *KMT2A*, *KMT2C*, *ARID1A*, *ARID1B*, *ARID2* displayed a comparable rate of mutations in both carcinomas and carcinoids. The mutations of the PI3K/AKT/mTOR pathway do not are frequent in neuroendocrine tumors, but their frequency of mutations is higher in carcinomas than in carcinoids [233]. Copy number alterations are more frequently observed in carcinomas than in carcinoids, such as focal deletion events involving *TP53* and *RB1* genes or focal gain events at the level of genes, such as *TERT*, *SDHA*, *RICTOR* and *PIK3CA* genes [233]. Interestingly, ACs displayed a hybrid pattern, with gains of *TERT*, *SDHA*, *RICTOR*, *PIK3CA*, *MYCL* and *SRC* at frequencies similar to those observed in carcinomas, whereas the *MEN1* loss was found at a rate similar to that observed in TCs [233].

These results were confirmed and extended by Centonze et al. who reported the next generation sequencing analysis of 790 lung NENs, subdivided into the four major subtypes [234]. The mutational rate and somatic mutation per case increased from TCs to SCLCs: mutational rate (%), median value 2.17 in TCs, 3.85 in ACs, 4.22 in LCNECs and 4.55 in SCLCs; somatic coding mutation per case, median value 1 in TCs, 2 in ACs, 3 in LCNECs and 7 in SCLCs [233]. Furthermore, the variability of these two parameters was much higher in carcinomas than in carcinoids. The genes most recurrently mutated in ACs were *EIF1AX* (4.8%). *ARID1A* (4.7%), *LPR1B* (4.3%), *NF1* (3.5%); the most frequently mutated genes in ACs were: *MEN1* (24.7%), *ATP1A2* (18.2%), *EIF1AX* (16.7%), *SPHKAP* 12.8%), *ARID1A* (9.6%) and *SMARCA4* (9.6%); mutations in tumor suppressor genes *TP53* and *RB1* were 0.8% and 1% in TCs and 5.3% and 2.2% in ACs [234]. It is important to note that *EIF1AX* gene mutations were enriched in both NETs but completely absent in NECs; *ARID1A* gene mutations were observed both in NETs and NECs but preferentially in ACs and LCNEC; *MEN1* gene mutations were much more frequent in ACs compared to TCs [234].

Laddha et al. have explored 30 well-differentiated NETs (17 TCs and 13 ACs) and observed that the most recurrently mutated genes were *MEN1* (13.8%), *ARID1A* (10%), *KTM2A* (3%), *KMT2C* (7%), *KMT2D* (3%) and *SMARCA4* (3%) [235]. Transcriptome and methylome profiles showed three distinct subtypes (LC1, LC2 and LC3): LC1 was characterized by the presence of *KTM2C*, *ARID1A* and *EIF1AX* mutations, high ASCL1 expression, high expression of cell cycle and mitotic genes expression; LC2 was characterized by the

presence of *MEN1* mutations, positivity of S100, expression of HNF1A and FOXA3; LC3 expression was characterized by the absence of mutations of histone modifier genes, absence of ASCL1 and S100 expression and endobronchial localization [235].

Alcala et al. have performed integrative and comparative genomic analyses on 171 pulmonary carcinoids, 75 LCNECs and 66 SCLCs. This integrative analysis allowed to stratify ACs into two prognostically relevant subgroups with significantly different 10-year overall survival, corresponding to 88% and 27%, respectively [236]. It was identified also a third group of tumors with carcinoid-like morphology but molecular profile more similar to that of LCNECs and defined as supracarcinoids; these observations support the hypothesis of a possible link between carcinoids and LCNECs, thus suggesting that a part of carcinoids may evolve into LCNECs [236]. Using the machine learning techniques and multi-omics factor analyses, 3 groups of lung carcinoids were identified. The first group (cluster A1) showed overexpression of ASCL1 and DLL3, high infiltration by dendritic cells; the second group (cluster A2) was characterized by recurrent mutations of *EIF1AX* and downregulation of the *SLIT1* and *ROBO1* genes; the third group (cluster B) was characterized by recurrent *MEN1* mutations, enrichment in monocytes and depletion in dendritic cells [236]. Among these three subgroups, the subgroup B had the worst median survival.

Genetic alterations occurring in LCNECs were extensively characterized in additional studies. LCNECs are distinguished from SCLCs based on various histologic criteria, including the cell size that is larger in LCNECs than in SCLCs, a polygonal cell shape, abundant cytoplasm and the presence of prominent nucleoli. Miyoshi et al. have performed a targeted capture sequence analysis of 244 cancer-related gene in 78 LCNEC tumor samples and compared the genetic alterations observed in these tumors with those of 145 SCLC samples [237]. At the mutational level the most frequent alterations were observed at the level of *TP53* (71%) and *RB1* (26%) genes; frequent alterations were observed at the level of various members of the PI3k/AKT/mTOR pathway (*RICTOR* 5%, *PIK3CA* 3%, *PTEN* 4%, *AKT2* 4% and *mTOR* 1%); mutations at the level of RTK pathway are observed in 14% of cases (*FGFR1* 5%, *KIT* 4%, *ERBB2* 4% and *EGFR* 1%); MAPK/ERK pathway alterations were observed in 11% of cases (*KRAS* 6%, *NF1* 4% and *HRAS* 1%); MYC family genes were amplified in 14% of cases (*MYCL1* 10%, *MYC* 3%, *MYCN* 1%) [237]. According to these findings it was concluded that LCNECs have a similar genomic profile to SCLC [237]. However, this study also evidenced some remarkable differences between LCNECs and SCLCs: the frequency of *RB1* alterations is lower in LCNECs than in SCLCs; the mutations of *LAMA1*, *PCLO*, *MEGF8* and *RICTOR* genes were more frequent in LCNECs than in SCLCs; copy number gains or amplifications involving *ERBB2* and *SETBP1* genes were more frequent in LCNECs than in SCLCs [237].

Rethkman et al. have reported the results of targeted next generation sequencing of 45 LCNEC using the MSK-IMPACT platform in association with a comprehensive clinic-pathologic evaluation [238]. The genes most frequently altered in these tumors were *TP53* (78%), *RB1* (38%), *STK11* (33%), *KEAP1* (31%) and *KRAS* (22%). According to the genomic profiles LCNECs were subdivided into two major and one minor subset: SCLC-like (40% of cases), characterized by *TP53* + *RB1* co-mutation/loss and *MYC*, *MYCL* and *SOX2* amplification; NSCLC-like (55% of cases), characterized by the lack of co-altered *TP53/RB1* and frequent NSCLC-like mutations, such as *STK11* (60%), *KRAS* (40%) and *KEAP1* (32%); carcinoid-like (5% of cases), characterized by *MEN1* mutation and low tumor burden [238]. In spite the similarities with lung adenocarcinoma, the NSCLC-like subtype exhibited peculiar genomic alterations, such as the frequent mutations in NOTCH family genes (28%) [238]. These subtypes differ also for some remarkable clinicopathologic properties, such as a higher proliferative activity in SCLC-like tumors and the presence of adenocarcinoma-specific differentiation markers in NSCLC-like tumors [238]. This study showed both the molecular heterogeneity of tumors classified as LCNECs and the existence of some difference between these tumors and SCLCs. A meta-analysis of all the main literature data on molecular landscape of LCNECs confirmed the existence of a SCLC-like subset, characterized by concomitant co-mutation of *TP53* and *RB1* [239]. These

observations have triggered the development of a prospective pilot clinical trial involving the treatment of LCNECs according to the *TP53 + RB1* co-mutation status.

Additional studies supported the existence of a subset of adenocarcinoma-like LCNEC and identified Napsin A as an immunohistochemical marker of these tumors. Napsin A (Novel aspartic proteinase of the pepsin family) is an enzyme involved in surfactant protein maturation, expressed in type II pneumocytes and currently used for the identification of lung adenocarcinomas. Napsin A was expressed in about 15% of LCNECs, at variance with lung carcinoids and SCLCs that were constantly negative [240]. Napsin A reactivity observed in LCNECs was lower compared to that reported in lung adenocarcinomas [239]. Napsin A LCNECs displayed the classical neuroendocrine morphology of these tumors and do not show a distinct adenocarcinoma component [240]. Genomic analysis showed that 78% of these tumors exhibited the presence of *KRAS* and/or *STK11* mutations [240].

Derks and coworkers have explored the possible influence of molecular heterogeneity on the response of LCNEC patients to chemotherapy [240]. The genomic profile of 78 LC-NEC patients was correlated with the response to two different chemotherapy regimens: a NSCLC chemotherapy based on the administration of platinum + gemcitabine + taxanes and a SCLC chemotherapy based on the administration of platinum + etoposide [241]. *RB1* mutation and protein loss were observed in 47% and 72% of these patients, respectively; patients with *RB1*-WT LCNEC treated with NSCLC chemotherapy displayed a longer median overall survival (9.6 months) than those treated with SCLC chemotherapy (5.8 months) [241]. The same outcome was observed for both patients with a *RB1* mutation or with Rb1 protein loss [241].

George and coworkers reported a comprehensive genomic and transcriptomic analysis of LCNECs [242]. LCNECs displayed an exome mutation rate of 8.6 mutations/Mb and a C:G>A:T transversion rate of 38.7%, suggestive of tobacco exposure [242]. This analysis confirmed the genetic heterogeneity of LCNECs and supported the existence of two molecular subgroups, comparable to those described above. Type I LCNECs were characterized by-allelic *TP53* and *STK11/KEAP1* alterations, *KRAS* mutations (less frequent that *STK11* or *KEAP1* mutations) *NKX2-1* amplification and *CDKN2A* deletion; type II LCNECs were characterized by bi-allelic inactivation of *TP53* and *RB1* [242]. Although type I LCNECs display some remarkable similarities at the level of genomic alterations with lung adenocarcinomas and squamous cell carcinomas, at gene expression level these tumors display a neuroendocrine profile, in part like SCLC; an opposite situation is observed for type II LCNECs, exhibiting similarities with SCLCs at the level of genomic alterations, but displaying a low neuroendocrine gene expression profile, with high activity of the NOTCH pathway [242]. At the level of copy number alterations, the most significant amplifications involved *MYCLN1* (12%), *NKX2-1* (10%), *FGFR1* (7%), *MYC* (5%) and *IRS2* (3%); the most significant deletions were observed at the level of *CDKN2A* (8%) and *PTPRD* (7%) [242]. The analysis of the mutational profile showed that 8 genes were frequently mutated in LCNECs. *TP53* was mutated in 92% of cases and *RB1* in 48% of cases; bi-allelic alterations in both these genes were observed in 40% of cases [242]. It is of interest to note that *RB1* alterations are preferentially observed in LCNECs exhibiting at histological level, admixtures with other histological components. After *TP53* and *RB1*, *STK11* (30%) and *KEAP1* (22%) are two frequently mutated genes; in most of the cases, *STK11* and *KEAP1* mutations do not co-occur; combined with loss of-of-heterozygosity, biallelic alterations of *STK11* and *KEAP1* are observed in 37% of cases [242]. The two genes encoding the metalloproteinases *ADAMTS2* (15%) and *ADMTS12* (20%) are frequently mutated, as well as the genes encoding *GAS7* (12%) and *NMT* (10%), not significantly mutated in other lung cancer types; the mutations affected functionally relevant protein domains, thus suggesting a potential role in LCNEC tumorigenesis [242]. RAS family genes (including *KRAS*, *NFE2L2* and *BRAF*) are altered either in consequence of mutational events or of focal amplifications in about 10% of cases; these findings indicate that SCLCs harbor also mutations of genes commonly mutated in lung adenocarcinomas but usually absent in neuroendocrine tumors [242]. The analysis of clonality of mutations showed that LCNECs exhibit only

a limited intratumor heterogeneity: only 7% of mutations were subclonal and all the main driver mutant genes have a clonal pattern [242]. The analysis of transcriptomic profiling showed that LCNEC tumors resembled more to SCLCs than to lung adenocarcinoma or squamous cell carcinomas. At the transcriptional level, two main types of LCNECs were identified: type I LCNECs display neuroendocrine profile, similar to SCLCs, characterized by ASCL1$^{high}$/DLL3$^{high}$/NOTCH$^{low}$ expression, whereas type II LCNECs display a low neuroendocrine profile, characterized by ASCL1$^{low}$/DLL3$^{low}$/NOTCH$^{high}$ expression, frequent *TP53* and *RB1* alterations and upregulation of immune-related gene pathways [242]. The ensemble of these findings supports the view that LCNECs constitute a distinct subset of HGNECs with peculiar histological and molecular properties and with a consistent degree of heterogeneity.

Simbolo and coworkers have explored the gene expression profiling of a series of ACs and LCNECs and have identified in these tumors three transcriptomic subtypes with specific genomic alterations: cluster 1 almost exclusively composed of LCNECs, displaying concurrent *TP53* and *RB1* inactivation in the absence of *MEN1* mutations; cluster 2 comprising a majority of ACs and a minority of LCNECs and displaying intermediate molecular abnormalities, including inactivation of *TP53* (40.9%), *MEN1* (22.7%) and *RB1* (18.2%); cluster 3 is composed by 85% of ACs and is characterized by recurrent *MEN1* mutations (37.5%) and less frequent *TP53* mutations (16.7%) [243]. As expected, patients in cluster 1 had a shorter overall survival than did patients pertaining to clusters 2 and 3 [242]. The intermediate pattern of C2 suggests a progression of malignancy for a part of ACs to LCNECs [243].

Zhuo et al. have explored the prognostic impact of genomic classification of LCNECs [244]. In this study the authors have shown also that the evaluation of mutational landscape of recurrently mutated genes using cell-free plasmatic DNA closely resembled that from tumor DNA [244]. Tumors displaying mutations/copy number loss of both *TP53* and *RB1* were classified as SCLC-like while all the rest was classified as NSCLC: patients with SCLC-like LCNEC have a shorter overall survival than those with NSCLC-like LCNEC despite higher response rate to chemotherapy [244].

LCNECs can be subdivided into pure LCNEC and combined LCNECs (C-LCNEC); the C-LCNEC is defined as a LCNEC type that is mixed with other histological components, such as adenocarcinoma or squamous cell carcinoma. The frequency of C-LCNEC varied from about 10% to 30% in different studies [245,246]. Milione et al. recently explored a cohort of 111 LCNEC patients, subdivided into a subgroup of C-LCNECs (31.5%), pure LCNECs (63%) and Napsin A-positive LCNECs (3.5%) [247]. A Ki-67 index cutoff of 55 was selected as predictor of overall survival in these patients, with about 30% of patients with a Ki-67 index <55% and 70% with a Ki-67 index >55% [247]. C-LCNECs exhibited a better overall survival compared to pure LCNECs; C-LCNECs with an adenocarcinoma component and Napsin A-positive LCNECs displayed a better overall survival than pure LCNECs and LCNECs with a squamous cell component [247].

Although the proposed classification of lung neuroendocrine tumors is able to provide an unequivocal definition of most of cases, some tumors, such as highly proliferative carcinoids, display in the same tumor the properties of tumors classified into two different types. Thus, these tumors, in spite a well differentiated morphology typical of carcinoids display a high mitotic index. Several studies have reported the properties of these highly proliferative lung carcinoids. Quinn et al. reported the characterization of 12 cases, showing the typical histological and cytomorphologic features of lung carcinoids, absence of anaplastic cytologic and of necrotic areas; most of these tumors displayed properties more in common with carcinoids than with LCNECs [248]. Rekhtman et al. have explored a group of 66 stage IV lung carcinoid and observed that 27% of sampled displayed mitotic counts higher than those expected for this type of patients; these high-proliferative cases displayed a well differentiated morphology, at genetic level displayed absence of *TP53* and *RB1* mutations and had a median overall survival of 2.7 years [249]. For the ensemble of these properties, these tumors should be considered as a separate group, distinct from high-grade neuroendocrine

cancers [249]. Rubino et al. reported the retrospective analysis of 514 lung carcinoid and observed that 6% of these tumors displays a high proliferative index, as defined by mitotic count >10/2 mm$^2$ and a Ki-67 index ≥20% [250]. In the whole population of patients with lung carcinoids, those with a Ki67 index displayed a median OS of 203 months, those with a Ki67 index of 6–20% showed an OS of 101 months and in those with KI67 index >20% the median OS was 53 months [250]. Hermans et al. explored 7 patients with stage IV, high proliferative lung carcinoids: 4/7 patients displayed preserved RB1 expression and exhibited a median overall survival of 45 months, while 3/7 patients showed decreased RB1 expression and a much lower median overall survival [251]. In conclusion, these studies support the view that lung carcinoids with a high proliferation index could be considered as a peculiar subset of lung neuroendocrine tumors more resembling to carcinoids than to LCNECs, with an intermediate prognosis between carcinoids and LCNECs.

It was suggested that lung NETs are separate entities as opposed to lung NECs. However, growing evidence suggests that at least a part of high-grade NENs of the lung may develop from pre-existing carcinoids. Thus, Pelosi et al. have performed a two-way clustering analysis of next generation sequencing data on 148 NETs, subdivided into six different histology clusters. According to this analysis they have concluded that low-grade lung NETs may evolve into high-grade tumors through two different, smoke related pathways: one pathway was compatible with the hypothesis of the evolution of TCs to LCNECs, whereas the other pathway was in line with a possible evolution from ACs to SCLCs [252]. Recently, Cros et al. reported the molecular analysis of 11 patients with high-grade (Ki-67 index >20% and mitotic rate >10%) lung neuroendocrine tumors with a carcinoid morphology [253]. The genomic characterization of these tumors showed some relevant alterations: Losses at the level of chromosomes 11, 3 and 13 were frequently observed; targeted NGS identified in 2/11 patients' mutations of *TPP53*, *ATM*, *PTEN*, *RAD50*, *TSC2* genes and in 1/11 cases mutations of *TSC1*, RB1 and *ARID1A* genes [253]. Comparative spatio/temporal analyses supported the view that these tumors derive from clones of lower malignancy and were genetically heterogeneous with a carcinoid mutational background and the progressive acquisition of NEC-like alterations, such as *TP53/RB1* alterations during tumor progression [253]. According to this view it was proposed that lung NENs can be subdivided into three groups: (i) primary high-grade NENs representing the most frequent NENS (70–75%) arising de novo without detectable precursor lesions; (ii) secondary high-grade NENs, observed in 20–25% of pulmonary NENs, with variable morphology and frequently showing the presence of precursor lesions; (iii) low-grade NENs, representing about 5% of pulmonary NENs [254].

The relationship between LCNEC and SCLC is at a large extent unknown and is a matter of debate. Mouse models could provide an important experimental tool to analyze possible biological relationships between LCNECs and SCLCs. In this context, Lazaro et al. described murine models of high-grade neuroendocrine lung carcinomas generated by the loss of four tumor suppressors [255]. The results of these studies showed that the type of neuroendocrine tumors generated in mice is related to the type of cells transformed with the four mutated tumor suppressors (*RBL1*, *RB1*, *TP53*, *PTEN*). Using the conventional cytomegalovirus promoter that targets most lung cell types, LCNECs were the predominant tumors developed in mice; using the adenovirus Ad5-K5cre that targets the keratin K5 promoter and then proximal airway basal cells, SCLCs were the predominant tumors originated in mice [255]. Molecular and transcriptomic analyses of both these two models supported a marked similarity to human counterparts [255].

## 9. SCLC Transformation from NSCLC

In some instances, NSCLCs can undergo a process of cellular transformation from NSCLC to SCLC. This transformation process is usually observed in NSCLC patients undergoing treatment with EGFR-TKIs (3–10% of EGFR-mutant cases) and represents one of the mechanisms of resistance to these drugs [256]. SCLC transformation from NSCLC has been observed also in EGFR-wild type lung cancer, or during treatment of

anaplastic lymphoma kinase (ALK)-mutant NSCLCs with targeted therapy and PD-1/PD-L1 immunotherapy [256].

Various studies have reported the analysis of consistent numbers of EGFR-mutated lung cancer patients developing SCLC. In this context, fundamental was the study of Marcoux et al. reporting a group of 67 patients with *EGFR*-mutated SCLC: at the initial lung cancer diagnosis, 57 of these patients had NSCLC and 9 de novo SCLC or mixed histology; *EGFR* mutations included exon 19 deletion (69%), L858R mutation (25%) and other (6%). All the patients with initial NSCLC received one or more lines of EGFR TKIs [257]. In these patients the median time to SCLC transformation was 17.8 months; after SCLC transformation, the patients displayed a response to platinum-etoposide and taxanes, with a median overall survival of 10.9 months after transformation [256]. The genotyping of these patients showed that all maintained their founder EGFR mutation and 15 of 19 with *EGFR* T790M positivity become WT at transformation; recurrent mutations in these SCLC-transformed patients included *TP53* (79%), *RB1* (58%) and *PIK3CA* (27%) [256]. Ferrer et al. reported a study on 61 SCLC transformed from NSCLC, 47 *EGFR*-mutant and 13 non-*EGFR*-mutant cases. The median time to SCLC transformation was shorted in the *EGFR*-mutant than in the non-*EGFR*-mutant group (16 months vs. 26 months); both tumor groups were sensitive to platinum-etoposide regimens [258]. The median overall survival was 28 months in the *EGFR*-mutant group and 37 months in the non-*EGFR*-mutant group; however, the median survival time after SCLC transformation was similar in the two groups (10 months in the *EGFR*-mutant group vs. 9 months in the non-*EGFR*-mutant group) [258].

It remains highly debated whether the NSCLC to SCLC transformation is a real event of transformation of pre-existing NSCLC cells into SCLC cells or whether it originates from the coexistence in the initial tumor of both tumor types (mixed histology). A contribution to address this problem derives from histological and molecular genetic studies.

An initial study by Niederst et al. provided fist molecular analysis of the genetic and epigenetic alterations observed in 11 SCLC transformed *EGFR*-mutated NSCLC and showed that: (i) transformed SCLC RNA profiles mimic those observed in de novo SCLCs; (ii) DNA sequencing showed some genetic lesions specific to transformed SCLC compared to NSCL, such as constant loss of *RB1*; (iii) transformed SCLCs displayed loss of EGFR expression [259].

Oser et al. proposed that both NSCLC and SCLC have a common cell of origin and analyzed according to literature data the cellular and molecular relationship of lung adenocarcinoma to SCLC [260]. A first element to be considered is related to the occurrence of some SCLCs with combined histology: about 10% of SCLCs display a LCNEC component; 2–10% of SCLCs have a NSCLC histological component [260,261]. A second element is related to the development of resistance to EGFR-TKIs, an event usually occurring within 12–18 months and most frequently represented by a Thr790Met mutation in *EGFR*; in 5–14% of these patients, the mechanism of resistance is related to a transformation from NSCLC to SCLC [262,263]. A third element is related to the presence in a minority (3–4%) of de novo SCLC patients of *EGFR* alterations; a part of these patients were less strong smokers and a part of them had combined SCLC/adenocarcinoma histology [264,265]. According to all these findings, Oser et al. have proposed the hypothesis that alveolar type II cells could represent the cells responsible for the formation of both lung adenocarcinoma and SCLC; according to this hypothesis, the type of tumor generated by the malignant transformation of these cells would be dependent on the mutational status of key oncogenes and tumor suppressors [260].

Zhang et al. reported the next generation sequencing analysis of 10 combined SCLC and of 30 pure SCLC [266]. At the clinical level there no significant differences in the two groups; overall survival of combined SCLC patients was worse than that observed for pure SCLC [266]. *TP53* and *RB1* were the most frequently mutated genes in both combined SCLC (83% and 66%) and pure SCLC (88% and 63%) groups; however, less than 10% common mutations were found in the two groups of tumors [266]. The analysis of the

mutational profile of the SCLC and NSCLC components of individual combined SCLC showed the presence of more than 50% of common mutations [266].

Lee et al. performed a longitudinal sequencing study to explore the genomic profile of NSCLC patients undergoing transformation to SCLC; whole genome sequencing was carried out for 9 tumors derived from 4 patients at various time points [266]. The divergence of SCLC ancestors from the lung adenocarcinoma cells occurred before the first treatments with EGFR TKIs and the inactivation of both *TP53* and *RB1* occurred at the early lung adenocarcinoma stages [267]. Analysis of additional 76 lung adenocarcinoma patients treated with EGFR TKIs showed that inactivation of both *TP53* and *RB1* was markedly more frequent in the SCLC-transformed group than in the non-transformed group; lung adenocarcinoma patients with *TP53* and *RB1* inactivation have a markedly higher risk of SCLC transformation [266]. Ahn et al. reported the longitudinal analysis of six lung adenocarcinomas that showed transformation to SCLC: four of these patients were pure SCLCs and two combined SCLC and adenocarcinoma; at clinical level, four of these cases were EGFR-mutant tumors from non-smoking females who were treated with EGFR TKIs (in these patients, the original EGFR mutation was retained in the transformed SCLC tumors) and the remaining two were EGFR-WT lung adenocarcinomas [268].

More recent studies have investigated the properties of combined SCLCs at cellular and molecular level. Thus, Zhao et al. have analyzed 170 SCLC cases and reported that 10 cases displayed a NSCLC component (5 cases adenocarcinoma and 5 cases squamous lung carcinoma) [269]. No significant clinical or pathological differences between pure SCLC and combined SCLC were observed; combined SCLC was associated with decreased overall survival compared with pure SCLC; the histologic components of combined SCLCs showed a high concordance but also showed divergent genotypes [269]. These findings were considered supportive of the origin from a common precursor, acquiring oncogenic changes in combined SCLC [269]. Lei et al. reported the study of 181 patients with combined SCLC and observed that: 58% were mixed SCLC/LCNEC, 13.8% SCLC/Adenocarcinoma and 13.2% SCLC/squamous lung carcinoma; patients with SCLC/LNCEC had longer disease-free survival compared to other patients with combined SCLC [270].

Recent studies support the view that concurrent *TP53* and *RB1* alterations define a subset of *EGFR*-mutant lung cancers at risk for SCLC histologic transformation. Offin et al. reported the results of a retrospective analysis based on 4112 lung cancer patients: 21% of these patients were *EGFR*-mutated and about 1% displayed triple *EGFR/TP53/RB1*-mutant genotype; all these triple-mutated patients had metastatic disease [271]. A total of 9% of the triple-mutated patients displayed a SCLC histology at the initial diagnosis; of the patients triple *EGFR/TP53/RB1*-mutant with lung adenocarcinomas (39/43), 18% had SCLC transformation during their disease course, with a median time to transformation of 1.1 years [270]. Triple-mutant lung cancers displayed an enrichment in concurrent mutations in *ERBB2*, *AKT3*, *SOX17*, *PTEN* [271]. The triple-mutant lung cancer population had displayed a higher incidence of whole-genome doubling compared to NSCLC or SCLC, with a further enrichment in triple-mutant cancers that transformed to SCLC; activation-induced cytidine deaminase/apolipoprotein B mRNA editing enzyme, catalytic polypeptide-like mutation signature (AID/APOBEC) was enriched in triple-mutated cancers that transformed to SCLC [271].

Xie et al. have explored by whole exome sequencing 5 patients with lung adenocarcinomas undergoing transformation to SCLC [272]. The results obtained in this study showed that (i) after SCLC transformation, the mutational spectrum changed with decreased C>T and increased C>A; (ii) copy number variation (CNV) burden of SCLC-transformed tumors was considerably increased; and (iii) the extent of CNV burden reduced the time to SCLC transformation and was associated with a shorter overall survival after SCLC transformation [272]. CNV burden seems to represent an important determinant in the transformation of NSCLCs to SCLCs [272].

Multi-omics analysis, based on genomic, epigenomic, transcriptomic, and protein characterization of combined NSCLC/SCLC tumors and of pre-/post-transformation samples,

provided evidence that the neuroendocrine transformation is more related to transcription program changes than to genomic alterations [273]. The genomic context in which neuroendocrine transformation is favored is represented by 3p chromosome loss, an event frequently occurring in SCLC [273]. In line with these findings, previous studies have shown that at the level of lung neuroendocrine tumors, loss of heterozygosity at 3p14.2 is most frequently observed in SCLC [274]. These tumors were also characterized by increased expression of genes involved in PRC2 complex and PI3K/AKT and NOTCH pathways; importantly, PI3K/AKT pathway inhibition reduced tumor growth and delayed neuroendocrine transformation in a mouse model, PDX-derived, of neuroendocrine transformation [274].

## 10. Conclusions

The studies carried out in the last years have led to the identification of the peculiar genetic abnormalities that characterize SCLC and other neuroendocrine lung tumors. Analysis of human tumors and of animal models have led to a molecular classification of SCLCs based on the dominance of transcriptional regulators.

These progresses in the understanding of the molecular basis and the biology of SCLC do not have been translated into corresponding clinical progress. However, the recent introduction of immune checkpoint inhibitors into the treatment of SCLC patients has been associated with prolonged clinical benefit in a subset of patients. However, the benefit of ICIs in SCLC is more limited than in NSCLC, thus suggesting that a better understanding of SCLC phenotypes and the identification of predictive biomarkers may provide more rational criteria for patient selection. Furthermore, the improvements in the techniques of molecular analysis have led to the identification of subsets of patients that may benefit from treatment with targeted therapies. A better understanding of SCLC at molecular level and of its consistent heterogeneity and plasticity is fundamental for the identification of therapeutic vulnerabilities and for the development of a more efficacious therapeutic strategy.

**Author Contributions:** U.T., E.P. and G.C. have equally contributed to critical analysis of literature data, to preparation of the manuscript and to editing of the manuscript. All authors have read and agreed to the published version of the manuscript.

**Funding:** This research received no external funding.

**Conflicts of Interest:** The authors declare no conflict of interest.

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
