# Peer review of "Genomic and Gene Expression Studies Helped to Define the Heterogeneity of Small-Cell Lung Cancer and Other Lung Neuroendocrine Tumors and to Identify New Therapeutic Targets"

_onco, doi:10.3390/onco2030013_

Round 1
Reviewer 1 Report
Testa, Pelosi and Castelli present an improved and corrected version of the manuscript.
All the points raised have been addressed. Therefore, the manuscript should be published in the present form
Reviewer 2 Report
It's already improved. I had no further requests
This manuscript is a resubmission of an earlier submission. The following is a list of the peer review reports and author responses from that submission.
Round 1
Reviewer 1 Report
In this manuscript, the authors made a comprehensive molecular characterization of Small-cell lung cancer and provided detailed information on therapeutic strategies. This manuscript would be informative and useful in practice. However, it would be better to link all chapters together, like in a decision tree or a cartoon. Figure 2 is a kind of example, but it's not intuitional. By the way, Several minor points are listed below.
- For table 1, it would be better to list in one column instead of three column
- Please make sure the gene nomenclatures are consistent.
- Please make sure the citations are with high precision. For instance, the first paragraph of the introduction is from one review article. In addition to citing this article, the original literature should be also cited in the appropriate sentence.
Author Response
- It seems very difficult to link all chapters together; therefore, we believe that it is better the leave the original structure of the paper with a subdivision into chapters.
- The Table 1 was rearranged to improve its look, in line with the reviewer’s suggestions.
- Gene nomenclatures are consistent.
- In the first paragraph of the introduction the appropriate citations were now added.
Reviewer 2 Report
Testa, Pelosi and Castelli propose a comprehensive analysis of the genomic alterations found in SCLC and the therapeutic strategies currently used.
The proposed review is well written and the amount of papers and clinical studies included are of interest for readers. It is, therefore, suitable for publication after the authors address some minor points:
- In general the most important criticism of the presented papers is the focus on closed and past clinical trials compared with new and ongoing one. A table with the most interest ongoing clinical trials would be interesting
- In Fig 1 they showed the results of a NGS study performed by George at al in 2015. Later in the paper the authors present and discusses the results of other papers (with similar results, but not completely identical). Would be better to modify the graph to make a mean of all the studies presented to have a broader view of the genetic landscape.
- In Fig 2 in the second and third row they specify the % of each subgroup but not in the first row. Would be better to include the % for each classification.
- In line 557 they say that all SCLC have inactive RB1, in fig 1 RB1 is mutated in around 60% of the case.
- In section 6.10 immunotherapy it is easier to follow if the authors present first the ICI as first-line treatment and move on in order as line of treatment instead of starting with ICI use as third-line
- In my opinion section 6.13 should have a section per-se (7 in this case) since is not strictly speaking about treatment options
- In general, it is necessary a revision of typos and formatting errors (such as the repetitions in line 797-798 and other)
- Also English should be carefully checked (i.e. in epidemiology it is better to use “account for” instead of “correspond to” when discussing % of cases)
Author Response
- A table with the most potentially interesting ongoing trials is now included.
- The Fig.1 now reports the mean value observed in the main studies on the molecular characterization of SCLC.
- The Fig.2 was modified, as suggested.
- In section 6.10, in line with the suggestion of the reviewer, we have modified the order of presentation of clinical data using ICIs in SCLC.
- In line with the reviewer’s suggestion,section 6.13 was now considered as a section per-se, section 7.
- Typos and formatting errors have been corrected.
- English was carefully corrected.
Reviewer 3 Report
I read through the review article of Testa U, Pelosi E, and Castelli G. I would like to congratulate the enormous efforts of the authors to cover everything about lung neuroendocrine tumors. The amount of information included here is just overwhelming.
But it seems that the authors are just ignoring the readers and just listing what they want to say. I would like to propose to separate the manuscript into two or three and publish them separately. (SCLC / other NETs etc.…) The structure of the manuscript is super redundant. The title and its content are not matching. The abstract is not summarizing what is written. The aim or focus of this review article is not clear. This manuscript feels like a textbook of NETs. There seems to be confusion about genetic mutation and protein expression. The connection between genetic mutations and ICI treatment is not clear. Unfortunately, the genetic classification of SCLC or NETs is not succeeding compared with lung adenocarcinoma. So, after the huge efforts of reading the whole manuscript, I just felt emptiness.
Author Response
- To meet a specific request, the title of the manuscript was modified and now is: Genomic and gene expression studies helped to define the heterogeneity of small cell lung cancer and other lung neuroendocrine tumors and to identify new therapeutic targets. This title now largely corresponds to the content of the text.
- The abstract strictly summarizes the content of the manuscript.
- The main aim of the manuscript consists to describe the most recent developments in the molecular characterization and classification of lung neuroendocrine cancer.
- The few connections existing between genetic abnormalities and response to ICI treatments in SCLC patients are now analyzed.
- We agree that the genetic classification of SCLCs or NETs is not succeeding compared with lung adenocarcinoma. However, the classification of SCLCs according to transcription factor expression now provides a tool to identify subsets of SCLC patients amenable to different treatments. This topic is in evolution and certainly in future years there will be a further improvement in the definition of the heterogeneity of SCLCs. Concerning NETs, the studies on the molecular characterization of LCNECs have provided evidence that 2-3 different subsets of these patients were identified, amenable for the evaluation of different first-line and second-line treatments.